# LeafAI: Interpretable plant disease detection for edge computing

**Md Abdullah Al Kafi**[1], **Sumit Kumar Banshal**[2]*, **Raka Moni**[1], **Aulia Luqman Aziz**[3], **Mohammed Aljuaid**[4], **Rohit Bansal**[5]

**1** Department of Computer Science and Engineering, Daffodil International University, Dhaka, Bangladesh, **2** Department of Computer Science and Engineering, Alliance University, Karnataka, India, **3** Faculty of Administrative Science, Universitas Brawijaya, Kota Malang, Indonesia, **4** Department of Health Administration, College of Business Administration, King Saud University, Riyadh, Saudi Arabia, **5** Rockford College, Sydney, New South Wales, Australia

* sumitbansha06@gmail.com

## Abstract

In real-world agriculture, healthy plant leaves are significantly more common than diseased ones. This natural class imbalance presents challenges in automated plant disease detection, as analyzing each leaf with computationally intensive deep-learning models is problematic, leading to inefficiency and increased resource consumption. To tackle this challenge and promote sustainable AI solutions, this study presents an iterative, hybrid AI approach that boosts computational efficiency, interpretability, and scalability for real-time disease detection. This hybrid system operates in two stages: first, a lightweight traditional machine learning classifier performs binary classification to quickly separate and exclude healthy leaves, followed by a deep learning model (ResNet, DenseNet, MobileNet, and EfficientNet) that classifies the specific disease in the smaller group of diseased leaves. This two-stage method minimizes computational load while maintaining high classification accuracy. Additionally, this study uses Explainable AI (XAI) methods, particularly Gradient-weighted Class Activation Mapping (Grad-CAM), to generate heatmaps. These heatmaps highlight the image areas that most significantly influence the model's predictions, thereby improving transparency and refining the feature extraction process. The proposed hybrid model, comprising Logistic Regression and Mobilenetv3, offers up to 77.6% faster inference than conventional deep learning models with only about 3% accuracy loss. For a large-scale test of 1,227 images on an entry-level laptop, the hybrid model reduced the total inference time from 4,548 seconds to just 1,010.13 seconds, with minimal CPU load. By addressing class imbalance, optimizing inference efficiency, and incorporating explainable AI, this work contributes a scalable, sustainable, and trustworthy solution for plant disease detection in precision agriculture.

**Data availability statement:** https://data.mendeley.com/datasets/hxsnvwty3r/1.

**Funding:** We would like to extend our appreciation to King Saud University for funding this work through the Ongoing Research Funding program (ORF-2026-481), King Saud University, Riyadh, Saudi Arabia. The funder has took role in preparing the finalized draft during the first draft. Also during the revision, funder helped in revising the manuscript effectively.

**Competing interests:** The authors have declared that no competing interests exist.

# 1 Introduction

Agriculture is essential for human civilization, supplying food and nutrition to billions of people globally. The spread of crop disease is affecting agricultural productivity and harming the global food supply every year. Early identification and classification of crop diseases can reduce crop losses and support food security. A 2022 report from the Food and Agriculture Organization (FAO) estimated that the world population will reach 9.1 billion by 2050, requiring approximately 70% growth in food production to maintain a steady supply [1]. To support this, the expansion of crop production and the minimization of crop disease are essential. This makes developing an automated plant disease identification system crucial for ensuring long-term agricultural productivity. However, early recognition is often challenging due to limited laboratory facilities and specialized knowledge [2]. Plant pathologists and farmers typically rely on manual visual observations to detect and diagnose plant diseases, drawing on their experience [3]. Sometimes, it can be prejudiced, erroneous, and cumbersome due to the need for enormous manual checks; however, early diagnosis is the leading solution to sustain organic farming [4]. Nevertheless, this manual approach is often ineffective for large agricultural fields for several reasons, including poor communication between specialists and farmers, prolonged time requirements, and additional costs. Researchers are now turning to advanced technologies, such as computer vision, drone monitoring, and machine learning, to develop faster and more accurate disease detection methods [5–7]. While these technologies offer high accuracy and rapid results, many existing studies overlook affordable deployment options for farmers in low-resource settings, as well as sustainability concerns such as excessive energy consumption, high hardware requirements, and additional costs. The expensive technology and infrastructure behind these systems make them inaccessible to many farmers in low-resource areas, where crop losses are most severe. As a result, these solutions often remain unrealistic for large-scale use, especially in developing regions with limited access to modern technology.

To address these challenges, this study presents an iterative, transparent, and explainability-focused hybrid AI approach. The primary goal of this approach is to enhance computational efficiency, make the models more transparent and understandable, and ensure they are scalable in real-world agricultural environments, ultimately supporting sustainable AI solutions in farming. Sustainable AI ensures the use of computational resources while maintaining high-level model performance. These make the AI systems more accessible, scalable, and environmentally friendly. Moreover, in real-world agriculture, data distributions are mostly skewed (class imbalance). The proposed methodology leverages this challenge into an advantage by employing a two-step classification process that enhances efficiency while ensuring accurate and reliable results. This study presents and assesses a two-step approach: first, a lightweight classifier quickly detects and filters out diseased leaves, followed by a deep learning model for precise disease classification.

The proposed architecture is scalable, and to evaluate its effectiveness, a mango leaves-based dataset has been assessed in this work. Additionally, the hybrid model is designed for flexibility, enabling it to be applied in diverse agricultural systems

while supporting scalable and sustainable disease detection. The proposed architecture addresses the need for computationally efficient, accurate, scalable, and sustainable solutions in crop disease detection deployed on edge devices. Furthermore, the presented efficient architecture fuels the advancement of the United Nations' Sustainable Development Goals (SDG 2 and SDG 12) by promoting more sustainable farming practices and improving food systems worldwide, focusing on commodity hardware and comparatively lower resource consumption in machine learning and deep learning deployment [8,9].

## 2 Related work

Modern agriculture plays a crucial role in ensuring food security and meeting the world's growing nutritional needs. Nowadays, it focuses more on sustainable and environmentally friendly production methods. Plant diseases are particularly prevalent in tropical climates, often resulting in significant crop losses and decreased agricultural productivity, especially in developing countries where farmers typically have limited income and access to technology [10]. Due to the contagious nature of plant diseases, early detection is essential for maintaining a steady food supply and safeguarding farmers' livelihoods by preventing substantial crop losses. For these shortcomings, accurately and promptly identifying plant diseases early is one of the main challenges of modern agriculture. To solve these issues, automated early detection systems can help reduce the reliance on excessive chemical use and lower production costs, promoting better environmental health and sustainable farming practices [11].

Traditional ways of finding plant diseases depend significantly on visual recognition and manual inspection, in which trained professionals look at crops for signs of disease [10]. Although these traditional methods can be effective in some instances, they are time-consuming and labor-intensive. In large-scale agricultural settings, this can make timely disease management more difficult. As agricultural operations expand, manual inspection becomes increasingly impractical. Additionally, visually inspecting subsets of leaves from trees can be brutal and prosaic [12]. These emphasize the pressing need for precise, low-cost, automated disease detection systems. To tackle these difficulties, technology, specifically deep learning, has a revolutionary impact on contemporary agricultural methods [13]. Deep learning, a subset of artificial intelligence, has introduced powerful tools for processing and analyzing large volumes of farming data. Unlike traditional machine learning, which depends heavily on hand-crafted features and domain-specific knowledge, deep learning models can automatically extract meaningful features directly from raw data [14]. This ability to identify representations makes deep learning particularly effective for various classification tasks, such as distinguishing between plant diseases based on leaf images. By capturing complex patterns and subtle visual cues, deep learning enables more accurate and scalable disease detection, offering a practical solution to the limitations of manual inspection [15].

Several studies have investigated deep learning approaches for plant disease detection, with many relying heavily on the "PlantVillage" dataset [14,16,17]. While this dataset has been instrumental in advancing early research, it is often criticized for being overly simplistic and lacking the diversity and variability of real-world agricultural settings [15,18,19]. To address these shortcomings, researchers have emphasized the importance of developing more authentic, field-collected datasets and have proposed enhancing deep learning performance by integrating hyperspectral or multispectral imaging [20–23]. These strategies aim to improve generalization and early detection, particularly under complex environmental conditions. The objectives align closely with the present study's focus on real-world performance and practical deployment [24,25].

In addition to improving dataset quality, scalability has been a central concern in plant disease detection research. For instance, a transfer learning approach using a pre-trained AlexNet model achieved a notable accuracy of 99.35% on a large-scale dataset containing 54,306 images, successfully classifying 26 different diseases across 14 crop species [14]. Similarly, a study utilizing the EfficientNetV2S architecture for apple leaf disease classification reached 99.21% accuracy by addressing class imbalance through runtime data augmentation [26].In another example, the InceptionResNetV2 model, trained on augmented grape leaf images from the PlantVillage dataset, achieved a training accuracy of 98.4%, a

validation accuracy of 99.5%, and a testing accuracy of 99.7%, underscoring its precision in disease detection [27]. This highlights the potential of deep learning to handle various classification tasks in agriculture when supported by robust training data. The current work builds on this foundation by applying deep learning models in a hybrid architecture that not only maintains high accuracy but also optimizes for speed and efficiency.

Beyond the architecture, recent work has focused on enhancing model performance through advanced preprocessing and feature extraction methods. For example, several studies have explored the integration of metaheuristic algorithms with convolutional neural networks (CNNs), following a structured pipeline that includes preprocessing, segmentation, feature extraction, and classification [28–30]. In the context of mango leaf disease detection, enhancement techniques such as contrast adjustment, fuzzy c-means clustering, geometric mean-based neutrosophic filtering, and histogram equalization have been employed to improve image quality and boost model accuracy [14,31]. These enhancements provide a robust foundation for CNN-based approaches, which continue to yield promising results in plant disease detection when combined with domain-specific strategies.

CNN-based approaches, combined with enhanced preprocessing techniques, have been explored in various disease detection studies. For example, early-stage identification of diseases in mango leaves, such as Anthracnose, Gall Midge, and Powdery Mildew, showed promising results using an Artificial Neural Network (ANN), which outperformed popular CNN models like ResNet-50, VGG16, and AlexNet [11]. Similarly, the LeafNet model demonstrated high proficiency in classifying seven major mango leaf diseases, achieving an average accuracy of 98.55%. Trained on a diverse image dataset collected from real-world agricultural fields in Bangladesh, LeafNet outperformed conventional CNN architectures, including AlexNet and VGG16, further validating the importance of tailored CNN design and domain-specific datasets for reliable disease classification [10]. Even conventional models, such as AlexNet, have proven effective when applied appropriately in agricultural settings. In one study involving 4,004 images of mango and potato leaves, AlexNet outperformed several standard CNNs in disease classification [32]. These findings reinforce the potential of carefully selected architectures for precise disease classification in agricultural settings. However, despite leveraging modern preprocessing techniques and transfer learning to address various shortcomings, these studies did not report model scalability, inference time, applicability, or key factors for real-world deployment.

Additionally, these models often lack interpretability, which can lead to trust issues when applied across various domains. In recent years, there has been a growing interest in integrating Explainable AI (XAI) into deep learning methodologies. XAI techniques enhance the transparency and interpretability of complex models, enabling researchers and practitioners to understand the decision-making processes of AI systems [33]. In agriculture, explainability is especially important because it helps build trust in AI predictions. When farmers and agronomists understand why a model made a particular decision, they are more likely to adopt and rely on these technologies for disease management. Different explainable AI methods are available for various tasks, such as Lime, Grad-CAM (Gradient-weighted Class Activation Mapping), and SHAP, which are among the most common methods [34].

In recent years, Grad-CAM (Gradient-weighted Class Activation Mapping) has emerged as a crucial tool in explainable AI, providing visual explanations for the decisions made by convolutional neural networks (CNNs). Grad-CAM offers valuable insights into the interior workings of deep learning models by emphasizing the regions of an input image that are most influential in a model's prediction. Grad-CAM was developed to address the interpretability challenges presented by deep learning models, particularly in image classification tasks [35]. Its utility has been demonstrated in various domains, including medical imaging, where it assists in identifying the regions of scans most relevant to diagnoses, as evidenced by subsequent studies. For example, Grad-CAM has been employed to visualize the features in chest X-rays that are instrumental in detecting pneumonia, improving the transparency and trust of AI-driven diagnostic tools [36]. Likewise, it has been implemented in agriculture to enhance the interpretability of models that detect plant diseases in multiple studies. Grad-CAM was employed to identify the critical areas of leaf images impacted by diseases such as blight and mildew [37]. This approach facilitated the identification of diseases with greater accuracy and transparency [38,39]. The technique's capacity to generate heatmaps that visually illustrate the significance of various regions of an image renders it as a potent

 

instrument for improving and debugging deep-learning models. Grad-CAM has also been integrated into educational platforms, helping students and researchers gain insights into how models interpret visual data [37,38,40,41]. Interpretability tools, such as Grad-CAM, are indispensable as AI systems continue to be implemented in critical environments. They enhance the transparency and reliability of AI applications, laying the groundwork for their ethical and responsible use in society.

While model interpretability is key, deploying deep learning models in real-world agricultural settings requires computational efficiency. Edge deployment remains challenging, as deep models often demand significant processing power and memory. The Open Neural Network Exchange (ONNX) offers a practical solution to this issue [42]. Created by Facebook and Microsoft, ONNX is an open format that facilitates the conversion of models from popular frameworks, such as PyTorch and TensorFlow, into a version ready for fast and efficient deployment across various platforms [43]. When used with ONNX Runtime, it supports a range of hardware, including CPUs, GPUs, and edge devices, helping to speed up inference and reduce memory requirements [44,45]. It also supports techniques such as model quantization and graph optimization, which help reduce the model size and make it run faster on devices with limited resources [46]. Unlike running models in their original training environments, ONNX enables the deployment of models in a lightweight and flexible manner, with improved performance [45]. It has already been used in real-world applications such as smart home devices, mobile apps, and healthcare tools running on edge devices. As edge and mobile computing become more common, tools like ONNX are becoming essential for building AI systems that are both fast and efficient. This is why ONNX is a suitable choice for this study, which aims to develop a robust deep-learning system that performs effectively in real-world scenarios.

While progress has been made in detecting crop diseases, some areas still need attention. For instance, there hasn't been enough assessment to see how these models perform with real-world data. Additionally, while some studies mention using their methods in practice, the particulars of the devices have been overlooked, such as whether it was a CPU or a GPU, or the performance metrics.

To address the existing gaps, the proposed two-step hybrid architecture leverages GRAD-CAM insights to guide the preprocessing and model development. IN the first step, Grad-CAM is applied to identify which regions of the leaf image contribute most to disease prediction. This knowledge is then used to design targeted preprocessing techniques. The second step develops models optimized for both inference speed, benefiting from the refined preprocessing pipeline. Finally, ONNX integration supports lightweight deployment on edge devices. By using Grad-CAM not only for explainability but also as a feedback mechanism for preprocessing, the architecture achieves reliable, scalable, and sustainable plant disease detection.

## 3 Methodology

The methodology is framed based on the established Knowledge Discovery in Databases (KDD) process, offering a structured architecture for extracting valuable insights from the test case of the leaf dataset [23,47].

Fig 1 shows the general workflow diagram for the whole experimental workflow.

### 3.1 Data description

The dataset comprises 4,000 JPG-formatted images of mango leaves, each with a resolution of 240 x 320 pixels. The photos were collected from four mango plantations in Bangladesh: the Sher-e-Bangla Agricultural University orchard, the Jahangir Nagar University orchard, the Udaipur orchard, and the Itakhola village mango orchard [48]. This dataset is well-suited for binary classification tasks, which include discriminating between healthy and diseased leaves. Additionally, it is suitable for multi-class classification, which enables the separation between various disorders. A total of seven prevalent diseases that harm mango leaves are included in the dataset. These diseases are as follows: a. Anthracnose, b. Bacterial Canker, c. Cutting Weevil, d. Die Back, e. Gall Midge, f. Powdery Mildew, and g. Sooty Mould. There is a

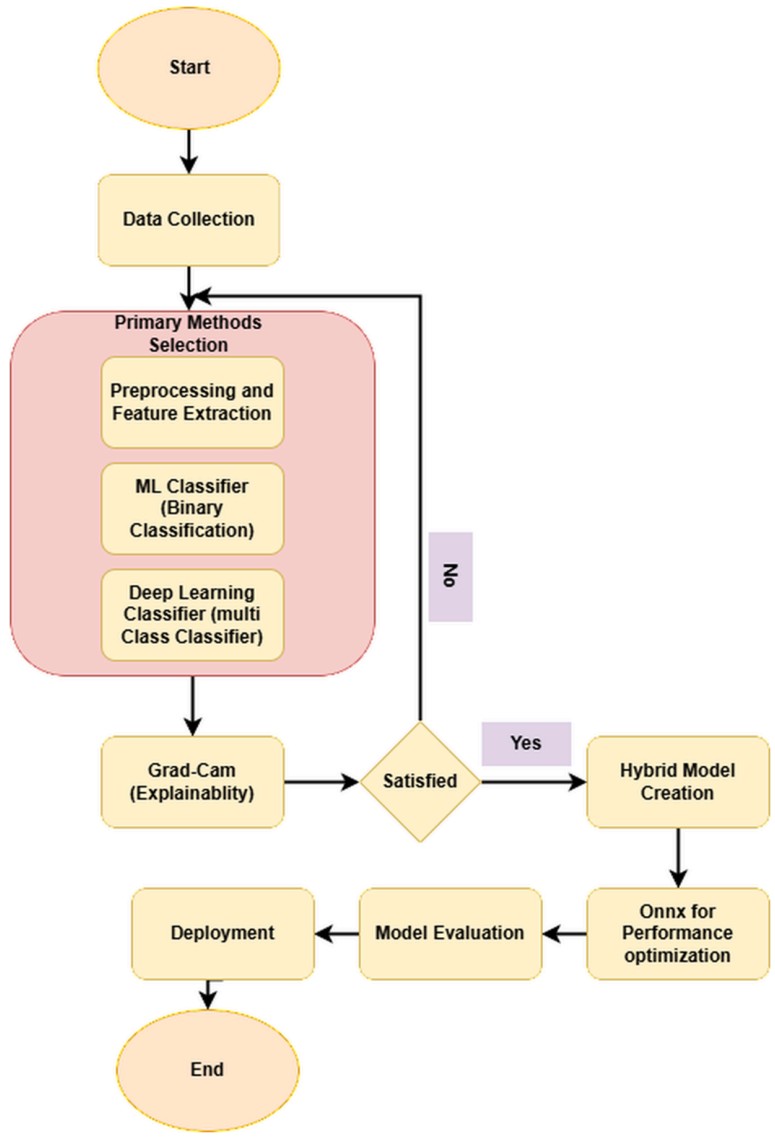

**Fig 1**. General workflow diagram.

category that represents healthy leaves, one of the eight defined classifications. The dataset includes eight classes: seven common mango leaf diseases and one healthy category. Anthracnose, caused by Colletotrichum gloeosporioides, affects foliage, blossoms, and fruits, often persisting post-harvest. Bacterial Canker leads to defoliation through angular lesions and spreads via rain or contaminated tools. Cutting Weevil (Hypomeces squamous or Sternochetus mangiferae) damages young shoots, causing twisted growth and dieback. Dieback results from fungal or bacterial infections and environmental stress, leading to branch withering. Gall Midge induces galls that deform leaves and stems, reducing yield and requiring integrated pest control. Powdery Mildew, caused by Erysiphe quercicola, appears as a white powder on plant surfaces, weakening photosynthesis. Sooty Mould, a secondary fungal issue from insect honeydew, blocks sunlight and disrupts photosynthesis. Lastly, healthy leaves are characterized by a vibrant green color, smooth edges, and well-defined vein structures. Fig 2 illustrates representative images for each class.

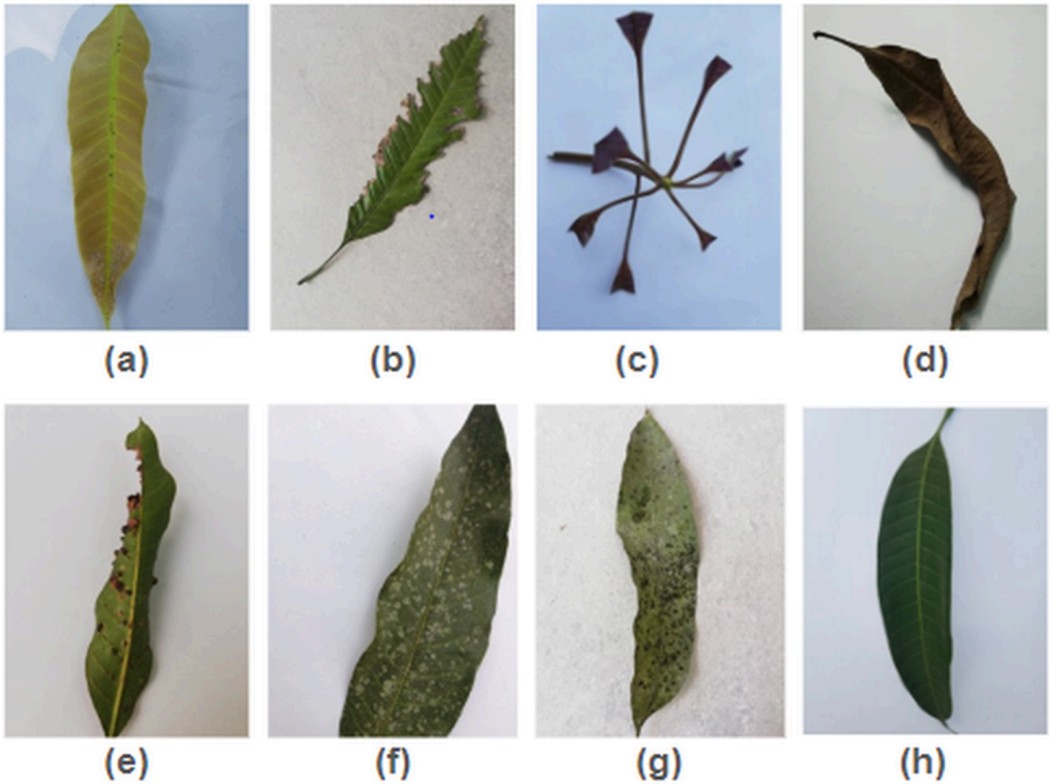

**Fig 2**. Sample images for each class ((a) Anthracnose, (b) Bacterial Canker, (c) Cutting Weevil, (d) Die Back, (e) Gall Midge, (f) Powdery Mildew, (g) Sooty Mould, (h) Healthy).

The dataset is split into 70%-15%-15% for training, validation, and testing. The training set comprises 2,800 images, with 600 reserved for validation and 600 for testing. For training, testing, and validation, having a balanced dataset is essential. Therefore, we will use a balanced dataset for these stages. To better simulate real-world inference, where data is often imbalanced, we will later experiment with mixtures containing different proportions of data in the following sections.

### 3.2 Experimental setup

The overview of the experimental setup for training and inferencing deep learning models in mango leaf classification is summarized in Table 1. This setup enhances computational efficiency, scalability, and real-world applicability, reflecting the research goal of advancing AI solutions in agriculture.

The training was conducted on a high-performance desktop equipped with a discrete GPU to handle the intensive demands of deep learning. However, to account for the limitations in rural and low-resource areas, the inference setup simulates a practical, lightweight laptop optimized for real-time deployment on energy-efficient hardware. This two-stage setup ensures the proposed hybrid model will remain accessible and adaptable. Making it deployable across various environments, from high-performance research labs to edge devices in remote agricultural areas.

The table provides a summary of the hardware specifications used for both training and inference. 8 GB RAM was used during inference on bthe oth the Desktop and laptops to mimic a real-world use case. Here on, the first laptop will be referred to as laptop one and the second laptop as laptop two.

**Table 1**. Hardware specifications for training and inferencing.

| Specification | Training Device | Inferencing Devices |
|---|---|---|
| CPU | Intel i5-13400 (65W) | Intel i5-12450H (45W, Laptop) |
| | | i5 8th gen 8365U(15 watt) |
| GPU | NVIDIA RTX 4070 (200W) | Intel Iris xe |
| | | Intel HD620 |
| OS | Manjaro Linux | Manjaro Linux (Laptop) |
| | | Manjaro Linux (Laptop) |

**Hardware Selection Rational:** The hardware was selected with two primary objectives: to ensure robust training performance and to facilitate practical, real-world deployment in low-resource agricultural environments. The training phase required significant computational resources to accelerate convergence and allow experimentation with large datasets and complex models. Therefore, a high-performance desktop equipped with a discrete GPU (NVIDIA RTX 4070) was utilized to facilitate an efficient training workflow.

For the inference phase, our focus shifted to simulating deployment conditions typical in rural and low-income farming regions. Two entry-level energy-efficient laptops with an Intel i5-12450H processor (45W TDP) and an i5-8365U (15W) were selected as the edge devices. This configuration strikes a balance between affordability, portability, and computational capability. It avoids reliance on cloud servers or expensive AI accelerators, ensuring the solution remains accessible and scalable.

The selected hardware reflects a realistic deployment scenario, where commodity computing devices — often the most viable option for rural users — are used to host AI solutions without sacrificing usability or performance. This approach aligns with the broader research goal of promoting inclusive and sustainable agricultural innovation.

**Training Setup:** Model training was performed on a desktop equipped with an Intel i5-13400 CPU and an NVIDIA RTX 4070 GPU, utilizing Manjaro Linux for efficient and stable development. To ensure balanced learning, training samples were selected to maintain class-wise uniformity across the dataset.

**Inferencing Setup:** Inference was executed on both laptops to simulate edge deployment. Real-time performance confirmed suitability for low-resource field use, with a consistent Linux environment ensuring smooth deployment. Only the test dataset was used for evaluation, with strict measures taken to prevent data contamination and leakage, and environmental variability.

### 3.3 Explainability and iterative refinement using grad-CAM

Understanding how decisions are made in deep learning models is often challenging due to their ```black-box``` nature. However, in high-stakes fields such as agriculture, transparency is crucial for building user trust and promoting adoption. To enhance interpretability, Gradient-weighted Class Activation Mapping (Grad-CAM) was employed to visualize the regions of mango leaves that most influenced the model's classification outcomes. These visual insights not only improve transparency but also inform architectural refinements.

Iteratively refining the model based on explainable AI techniques ensures that the architecture evolves to be more efficient and accurate. By highlighting key features contributing to disease classification, Grad-CAM guides feature extraction and model training refinement, making the system more adaptable for real-world agricultural applications [35]. This process aligns with the broader objective of creating AI systems that are efficient, scalable, interpretable, and trustworthy for agricultural stakeholders.

### 4 Preliminary model assessment

To establish a comprehensive performance baseline, we conducted an in-depth evaluation of four prominent CNN architectures: **ResNet**, **DenseNet**, **MobileNet**, and **EfficientNet**. This initial phase was crucial for determining the most

suitable models for our proposed hybrid approach, balancing high accuracy with the computational efficiency required for deployment on resource-limited devices. We assessed their performance across key metrics, including inference time, model size, and F1 score, and used statistical analysis to confirm the significance of our findings. The following subsection details these results.

## 4.1 Comparative analysis of CNN models

The initial comparative analysis of the four models is summarized in the Table 2. As shown, the **EfficientNet** model achieved the highest F1 score at 98.4%, demonstrating its superior accuracy in classifying plant diseases. The **MobileNet** model also performed exceptionally well, with an F1 score of 97.1%, while being the most computationally efficient. It had the smallest model size (8.9 MB) and the fastest inference time (86ms). In contrast, the ResNet and DenseNet models, although still accurate with F1 scores of 95.4% and 95.5%, respectively, were significantly larger and slower to make predictions. These findings highlight a clear trade-off between model complexity and performance, with newer, more optimized architectures, such as **EfficientNet** and **MobileNet**, offering a more favorable balance.

## 4.2 Cross-validated performance analysis

To further validate these findings, we conducted a more rigorous 10-fold cross-validation to assess the stability and reliability of each model's performance. The results are presented in Table 3 and visualized in Fig 3. The bar chart visually confirms that **EfficientNet** and **MobileNet** had the highest mean F1 scores, with more minor standard deviations compared to ResNet and DenseNet. The small 95% confidence intervals for MobileNet and EfficientNet indicate that their performance is consistently high across different data folds, demonstrating their robustness. In contrast, the larger confidence intervals for ResNet and DenseNet suggest more variability in their performance.

To determine if the performance differences between the models were statistically significant, a paired t-test was performed, with the results shown in Table 4 and Fig 4. A p-value less than 0.05 is generally considered statistically significant.

- **ResNet vs. EfficientNet:** The p-value was 0.022, which is less than 0.05. This means the performance difference between ResNet and EfficientNet is **statistically significant**, confirming that Efficient is a demonstrably better model.
- **Other comparisons:** All other pairwise comparisons (e.g., MobileNet vs. EfficientNet, DenseNet vs. MobileNet) showed p-values greater than 0.05, indicating that their performance differences were not statistically significant. This is an

**Table 2**. Comparative model performance metrics (Cross validation 5 fold).

| Model | Inference Time GPU(ms) | Size | F1 score |
|---|---|---|---|
| ResNet | 205 | 44.7mb | 95.4 |
| DenseNet | 194 | 29.2mb | 95.5 |
| MobileNet | 86 | 8.9mb | 97.1 |
| EfficientNet | 86 | 16.1mb | 98.4 |

**Table 3**. 10-Fold cross-validation performance summary for CNN models (GPU).

| Model | Mean F1 | Std | 95% CI |
|---|---|---|---|
| ResNet | 0.9544 | 0.0287 | [0.934, 0.975] |
| DenseNet | 0.9551 | 0.0484 | [0.920, 0.990] |
| MobileNet | 0.9709 | 0.0144 | [0.961, 0.981] |
| EfficientNet | 0.9836 | 0.0167 | [0.972, 0.996] |

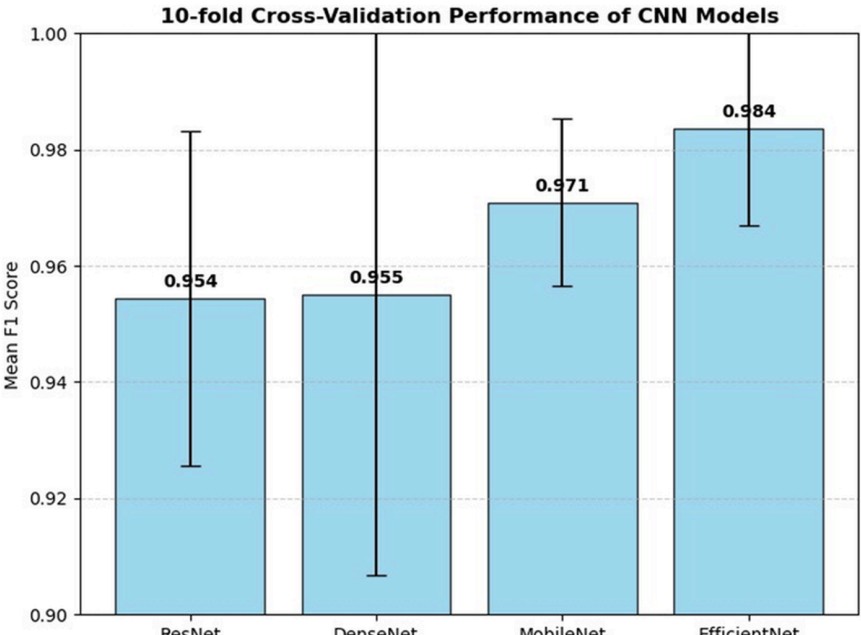

**Fig 3**. Deep learning model cross-validation result.

**Table 4**. Paired t-test results between CNN models.

| Comparison | t-statistic | p-value |
|---|---|---|
| ResNet vs DenseNet | −0.038 | 0.971 |
| ResNet vs MobileNet | −1.528 | 0.161 |
| ResNet vs EfficientNet | −2.753 | 0.022 |
| DenseNet vs MobileNet | −1.149 | 0.280 |
| DenseNet vs EfficientNet | −1.750 | 0.114 |
| MobileNet vs EfficientNet | −2.101 | 0.065 |

important finding, as it suggests that while EfficientNet had the highest F1 score, its performance advantage over MobileNet was not significant enough to rule out chance.

Additionally, Fig 5 shows a detailed view of the model's classification performance across all eight disease classes in the confusion matrix. The high values along the diagonal (e.g., 103 for Anthracnose and 100 for Powdery mildew) demonstrate the model's strong accuracy. The low numbers off the diagonal confirm that the model efficiency distinguishes between different diseases, with only minimal misclassifications occurring between a few classes. This result reinforces the model's reliability for multi-class plant disease detection.

In conclusion, these initial results indicate that while EfficientNet achieves the highest accuracy, MobileNet offers a compelling combination of high performance and model size, making it an excellent candidate for deployment on resource-constrained devices.

**4.2.1 Grad-Cam analysis.** Gradient-weighted Class Activation Mapping (Grad-CAM) was applied to gain deeper insights into the distinguishing features between healthy and diseased mango leaves, directly influencing the design of the hybrid pipeline. Grad-CAM is a widely used visualization technique in computer vision that addresses the interpretability challenges of convolutional neural networks (CNNs) [49]. Grad-CAM creates heatmaps that show which parts of the

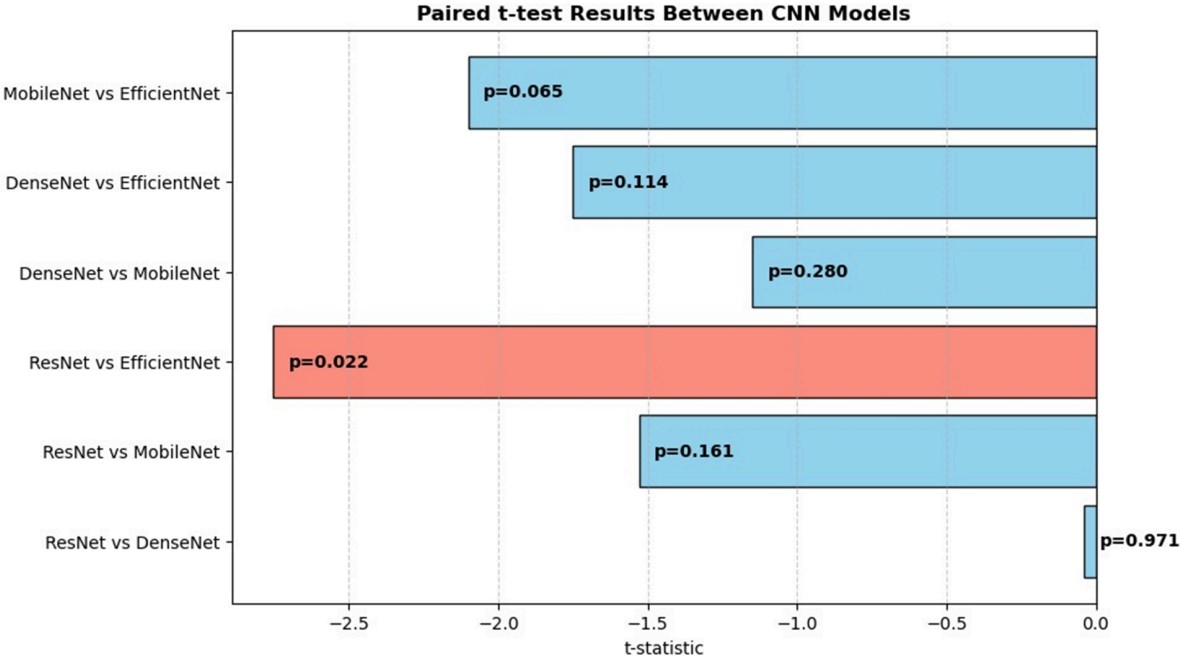

**Fig 4**. Deep learning model cross-validation result.

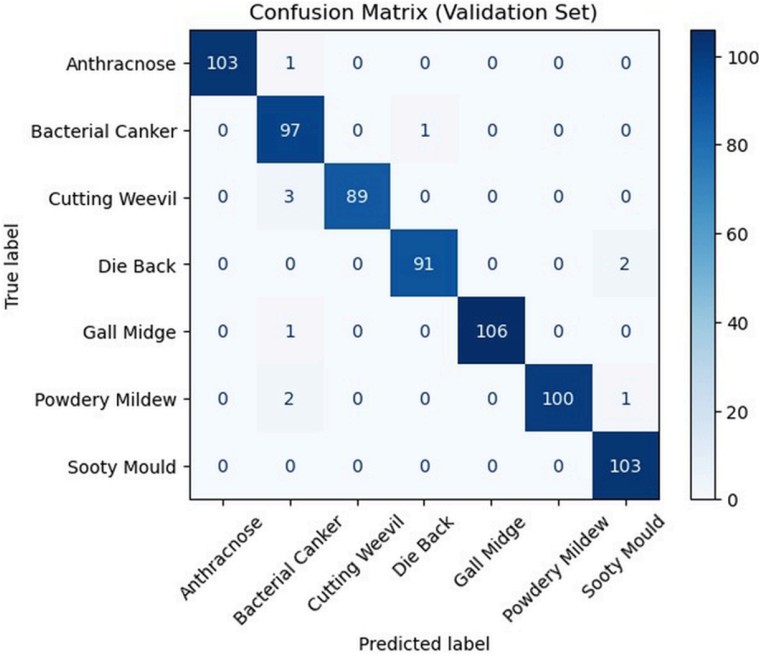

**Fig 5**. Confusion matrix MobileNet (initial).

image the model focuses on when making its classification decisions. This visualization enhances transparency and offers a more intuitive understanding of the model's prediction process, supporting informed refinement and increased trust in the system's outputs.

Grad-CAM operates through four key steps:

- **Forward Pass**: When the input image goes through the CNN, it first generates feature maps from the last convolutional layer, followed by class scores from the fully connected layers.
- **Gradient Calculation**: The gradients of the predicted class score are calculated with respect to the feature maps in the last convolutional layer.
- **Weight Calculation**: The gradients are averaged across all locations to generate weights that indicate how much each feature map contributes to the model's prediction.
- **Heatmap Generation**: The calculated weights compute a weighted sum of the feature maps, and a ReLU activation is applied to produce the final heatmap.

The heatmaps provide valuable interpretability by overlaying these regions on the original leaf images, illustrating where the model concentrates its attention when classifying leaves as healthy or diseased. Fig 6 visualizes sample heatmaps for healthy and unhealthy leaves.

**4.2.2 Analysis of healthy vs. diseased leaves.** The Grad-CAM analysis from Fig 6 offers valuable insights into how the model distinguishes between healthy and diseased leaves, which is crucial for developing a hybrid, explainable AI pipeline. As outlined in Table 5, the model's focus on specific leaf features highlights key structural and textural differences that guide accurate disease classification.

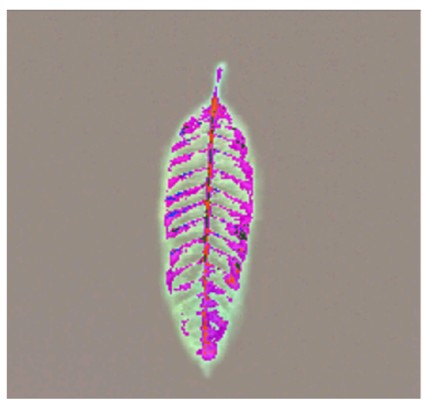
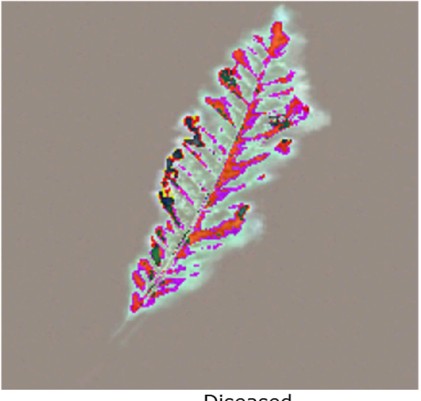

Healthy Diseased

**Fig 6**. **Grad-CAM illustrating the model's focus for healthy and diseased leaves.**

**Table 5**. **Comparative analysis of features for healthy and diseased leaves (based on Grad-CAM insights).**

| Feature | Healthy Leaf | Diseased Leaf |
|---|---|---|
| Color Highlights | Uniformly distributed around veins and edges | Irregular, intense highlights focused on damaged areas |
| Highlighted Regions | Vein structure, overall shape | Discoloration, lesions, edge damage |
| Model Focus | Uniform coloration, intact contours | Areas with visible damage, spots, anomalies |
| Vein Structure | Prominently defined and uniformly highlighted | Often less visible, except when damaged |
| Leaf Shape | Intact, undamaged | Irregular, with visible deterioration |
| Discoloration and Spots | Minimal or absent | Significant, attracting model focus |
| Edge Condition | Smooth, intact | Deteriorated, highlighting damaged edges |
| Heatmap Intensity | Moderate | High, emphasizing critical areas of concern |
| Structural Integrity | High, contributing to classification as healthy | Low, contributing to classification as diseased |

The Grad-CAM heatmaps highlight the veins and edges for healthy leaves, indicating that the model focuses on the leaf's overall shape and structure. It focuses on clear patterns, such as well-defined veins and smooth edges, which help it recognize a healthy leaf. The moderate intensity of the heatmap suggests healthy picks up on consistent features typical of healthy foliage. This insight is valuable for sustainable farming, as it facilitates the early detection of healthy crops, making it easier to manage them proactively without requiring constant manual checks.

On the other hand, diseased leaves show intense, irregular heatmap highlights in areas with visible damage, such as discoloration, spots, and edge deterioration. These highlights indicate the model's focus on unusual regions where the leaf's color, texture, and structure have been compromised. The heatmap intensity is more vigorous in these areas, highlighting anomalies such as irregular contours, less-defined veins, and more noticeable edge damage. All of these factors help the model identify the leaf as diseased. This insight is crucial for efficient disease detection in farming, as the AI can quickly and accurately spot early signs of disease, reducing waste and minimizing crop loss.

Table 5 below summarizes the key features of the practical model used to differentiate between healthy and diseased leaves, highlighting the importance of this explainability in practical agricultural applications. The insights gained from this analysis form the basis for refining our disease detection pipeline, ensuring it is adaptable and useful for deployment in various resource-limited environments.

Additionally, examining specific diseased leaves revealed that some share similar shapes and vein structures. This overlap suggests that relying solely on shape or vein-based preprocessing is insufficient to accurately differentiate between diseases. To overcome this limitation, disease-specific preprocessing techniques were introduced and integrated with deep learning models, enabling more precise and nuanced classification. By leveraging the feature-learning capabilities of deep networks, the system can capture subtle visual patterns that traditional preprocessing methods often overlook, thereby enhancing overall classification accuracy.

- **Anthracnose:** Gradcam view of Anthracnose in Fig 7 shows that the model focuses on specific parts of the image with spots and other anomalies.

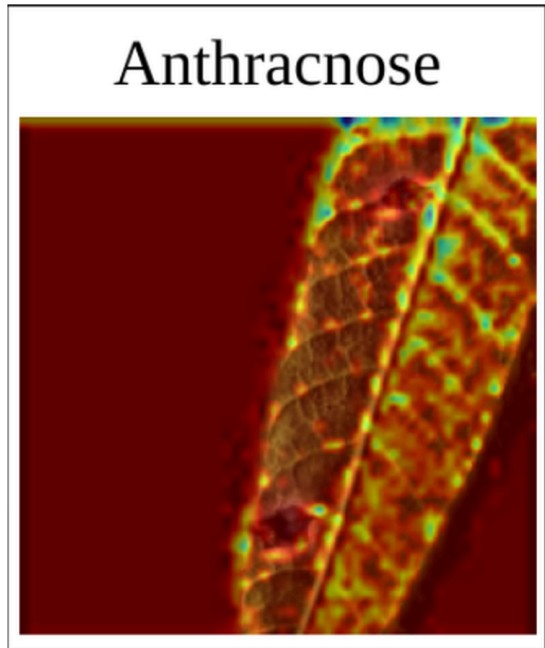

**Fig 7**. **Anthracnose Gradcam.**

- **Bacterial Canker:** Similar to Anthracnose, it can be seen in Fig 8 that the model focuses on specific features of the image in the Bacterial Canker photo. Such as the dead and discolored part of the photo, and dead spots.
- **Cutting Weevil:** Fig 9 shows that the model focuses on the small leaf left on the edge not on the entire structure of the leaf.

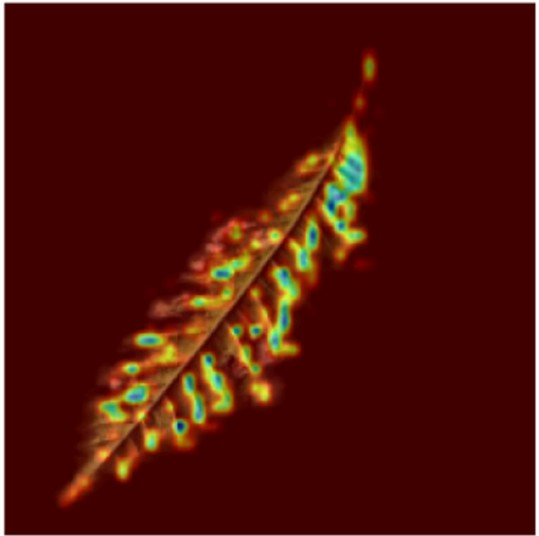

**Fig 8**. **Bacterial Canker Gradcam.**

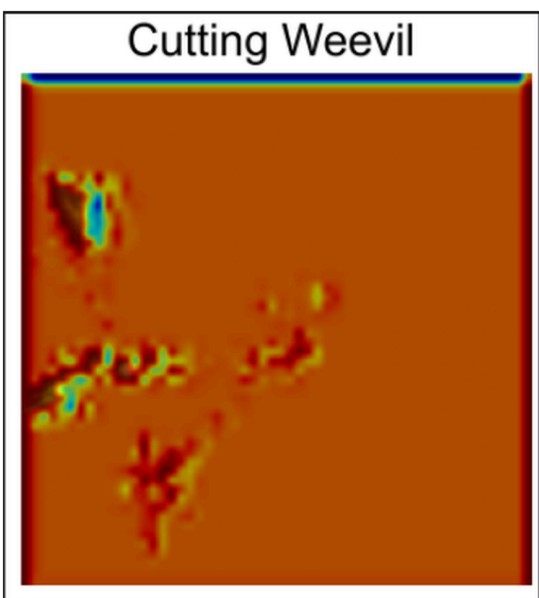

**Fig 9**. **Cutting Weevil Gradcam.**

- **Die Back:** In Fig 10, the Grad-cam analysis of Die Back also shows a different trend than others. The model focuses primarily on the rolled part of the image, rather than other features.
- **Gall Midge:** Fig 11 shows how the model focuses on how the leaf changed from one end to another end due to the disease. It focuses on both the disoriented and healthy parts of the leaf.

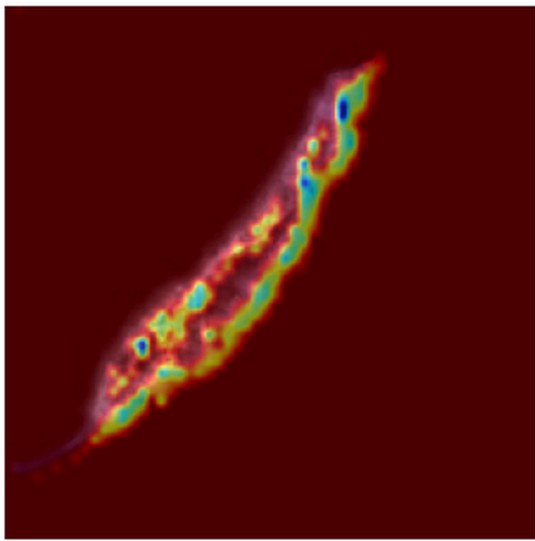

**Fig 10**. **Die back Gradcam.**

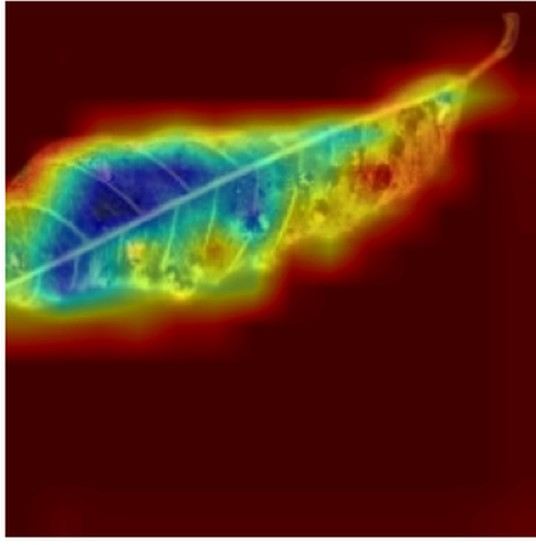

**Fig 11**. **Gall Midge Gradcam.**

- **Powdery Mildew:** The gradcam analysis of Powdery Mildew in Fig 12 shows that the model focuses on the spots and its orientation with density on the image only.
- **Sooty Mould:** In Fig 13, from gradcam analysis, it can be seen that the model focuses on the spot orientation of Sooty mould in the image.

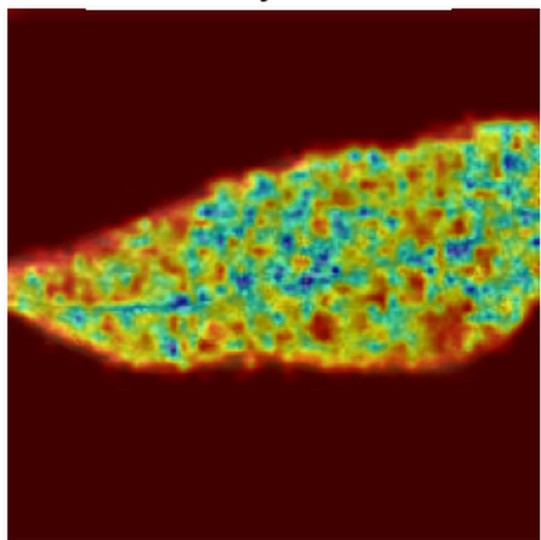

**Fig 12**. **Powdery Mildew Gradcam.**

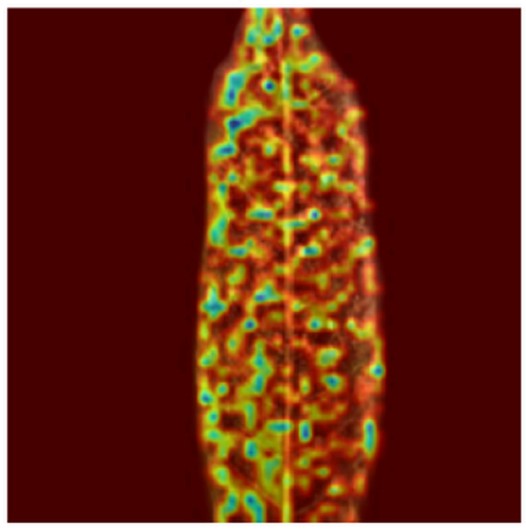

**Fig 13**. **Sooty Mould Gradcam.**

From the grad cam analysis, it is clear that the difference between healthy and diseased leaves is easier to determine; however, the distinction between all the diseased leaves is a complex process. Moreover, the model focuses on different parts of each image depending on the disease. Therefore, it is clear that a two-stage method would be a suitable choice to handle each task separately, thereby gaining a computational advantage.

**4.2.3 Grad-CAM-guided preprocessing strategy.** The insights from Grad-CAM played a crucial role in shaping the preprocessing strategy for the two-stage hybrid pipeline. In the first stage, healthy leaves can be identified by contour-based features such as vein structure and smoothness. Then, traditional feature extraction methods, such as Local Binary Patterns (LBP), Histogram of Oriented Gradients (HOG), and Hu Moments, are used, effectively capturing edges, textures, and contours.

The Grad-CAM analysis revealed that while diseased leaves share some similarities in shape and structure, healthy ones differ significantly in color, texture, and signs of damage. Based on this, enhancement techniques are employed in the second stage of this process to capture these subtle differences more effectively. By focusing on details such as color variations, texture, and damage areas, this study aimed to give the model a better chance of identifying the small yet significant signs of disease in leaves.

Overall, this Grad-CAM-driven preprocessing approach enhances the interpretability of the hybrid pipeline while ensuring that each classification stage focuses on the most relevant and biologically meaningful features. This biologically motivated design improves classification accuracy and optimizes computational efficiency, making the model practical and efficient for large-scale agricultural datasets.

## 4.3 Hybrid model architecture

To refine the two-stage hybrid pipeline, several classical machine learning (ML) models were first evaluated on the initial classification task of differentiating between healthy and diseased leaves. This step was crucial for selecting a lightweight yet effective classifier for the first stage of the pipeline, which handles the initial filtering of healthy leaves. By employing a simple ML model for this task, the computational load can be significantly reduced, increasing the overall efficiency of the system. The models evaluated were RandomForest, Support Vector Classifier (SVC), and Logistic Regression.

**4.3.1 Performance analysis of classical ML models.** The performance of these classical ML models was rigorously evaluated using 10-fold cross-validation, with the results summarized in Table 6 and visualized in Fig 14. The bar chart clearly shows that Logistic Regression achieved the highest mean F1 score of 0.919. This was followed by SVC (0.897) and RandomForest (0.878).

These results indicate that even simple models can effectively perform the initial classification task with high accuracy. The tests' F1 scores, also displayed in Table 6 and Fig 14, show a similar trend, with Logistic Regression again outperforming the other models.

To gain a granular understanding of each model's performance, the confusion matrices were analyzed, as shown in Fig 15. The matrices provide a breakdown of correct and incorrect predictions for both healthy and diseased leaves, offering insights into the strengths and weaknesses of each model. The analysis provides clear evidence for the model performance rankings. Logistic Regression, which achieved the highest F1 score, also demonstrates the most balanced and accurate predictions, with only eight healthy leaves misclassified as diseased (false positives) and five diseased leaves misclassified as healthy (false negatives). The SVC model also performed exceptionally well, with a remarkably

**Table 6**. 10-Fold cross-validation and test performance for classical ML models.

| Model | Mean F1 | Std | 95% CI (CV) | Test F1 |
|---|---|---|---|---|
| RandomForest | 0.8778 | 0.0436 | [0.847, 0.909] | 0.8848 |
| SVC | 0.8968 | 0.0362 | [0.871, 0.923] | 0.9247 |
| LogisticRegression | 0.9187 | 0.0303 | [0.897, 0.940] | 0.9350 |

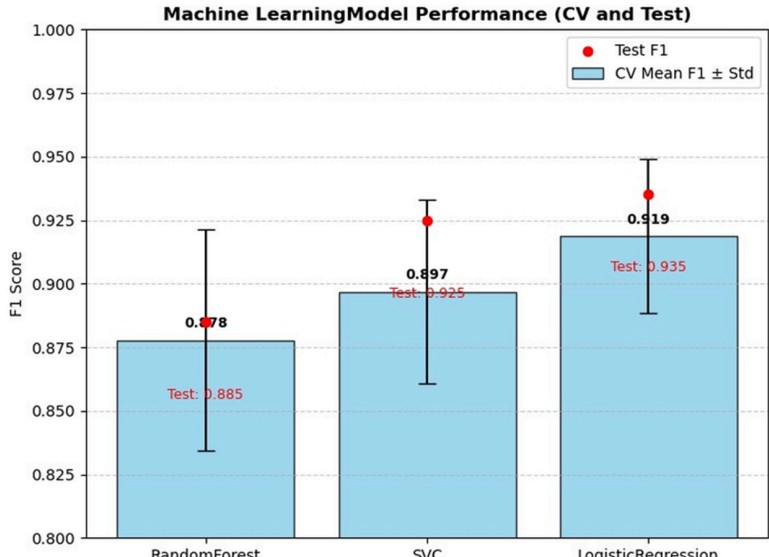

**Fig 14**. **Machine learning algorithms performance (cross validation).**

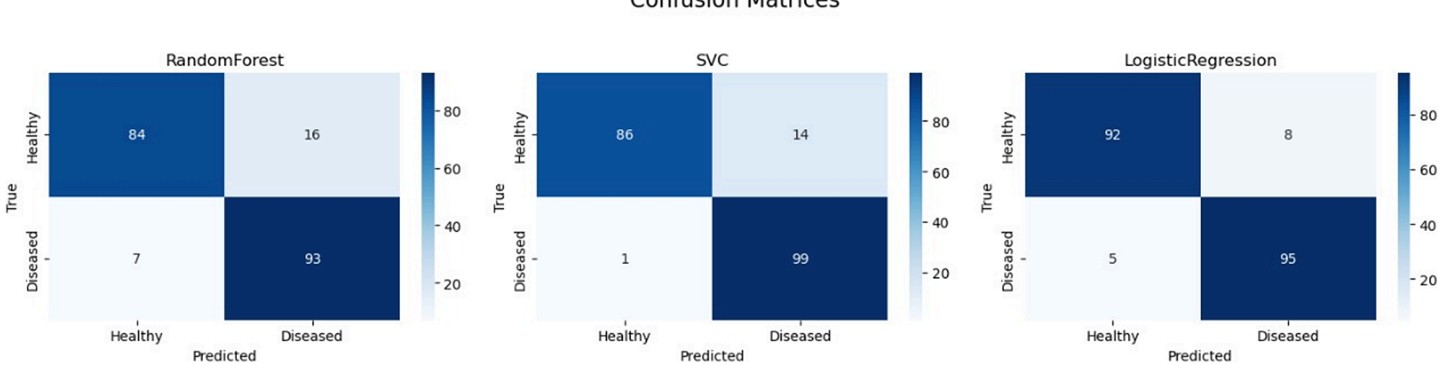

**Fig 15**. **Confusion matrices for the RandomForest, SVC, and logistic regression models.**

low number of false negatives (1). This indicates its strong ability to accurately identify diseased leaves. However, it made slightly more false positive errors (14) than Logistic Regression, classifying a healthy leaf as diseased. The Random Forest model, in contrast, shows a higher rate of misclassification in both categories (sixteen false positives and seven false negatives), which accounts for its lower overall F1 score. Collectively, the confusion matrices visually confirm that both Logistic Regression and SVC are highly reliable for the initial filtering stage.

**4.3.2 Statistical significance of ML model performance.** A paired t-test was performed to validate the observed performance differences between the classical ML models. As presented in Table 7, the p-value of 0.019 for the RandomForest vs. Logistic Regression comparison confirms that Logistic Regression holds a statistically significant performance advantage. This result solidifies its selection as the best model for the initial classification stage of the hybrid pipeline. The other comparisons enable a confident statement that one model is genuinely superior to another for a specific task, thereby necessitating that differences could be due to chance.

**Table 7. Paired t-test results between ML models.**

| Comparison | t-statistic | p-value |
|---|---|---|
| RandomForest vs SVC | −2.220 | 0.054 |
| RandomForest vs LogisticRegression | −2.847 | 0.019 |
| SVC vs LogisticRegression | −2.190 | 0.056 |

The purpose of this statistical analysis was to provide a rigorous, data-driven validation for the model selection process. The p-value enables a confident statement that one model is genuinely superior to another for a specific task, thereby avoiding conclusions based on minor, random performance fluctuations.

In conclusion, the analysis confirms that Logistic Regression is the most effective and reliable model for the initial classification of healthy vs. diseased leaves. Its strong performance and computational efficiency make it an ideal candidate for the first stage of the hybrid architecture, enabling the rapid and accurate filtering of healthy leaves and thereby reducing the computational demands on the subsequent deep learning stage. This two-stage approach ensures both high accuracy and sustainability for real-world agricultural deployment.

**4.3.3 Stage 1: Preprocessing for machine learning model.** The initial phase of the pipeline focuses on preprocessing tailored for the Random Forest classifier. This step ensures efficient feature extraction and normalization, allowing the RF model to handle input data more effectively. The preprocessing in this stage involves the following steps:

1. **Grayscale Conversion:**

    The original RGB image $I_{RGB}(x, y)$ is first converted into a grayscale image $I_{gray}(x, y)$. This step reduces computational overhead by discarding color information that is less critical for the RF-based feature extraction. The grayscale value is computed using the following weighted sum:

$$I_{gray}(x, y) = 0.2989 \cdot R(x, y) + 0.5870 \cdot G(x, y) + 0.1140 \cdot B(x, y), \tag{1}$$

    where $R(x,y)$, $G(x,y)$, and $B(x,y)$ are the red, green, and blue channel intensities, respectively.

2. **Min-Max Normalization:**

    We apply min-max normalization to the grayscale image to ensure the pixel intensity values fall within a uniform scale. This is expressed mathematically as:

$$I_{norm}(x, y) = \frac{I_{gray}(x, y) - \min(I_{gray})}{\max(I_{gray}) - \min(I_{gray})}, \tag{2}$$

    where $I_{norm}$ represents the normalized grayscale image, and the pixel intensities are mapped to the range [0,1]. This normalization improves the RF classifier's ability to distinguish patterns by reducing pixel intensity biases.

After preprocessing, the normalized grayscale image $I_{norm}$ is passed to the Random Forest classifier, which extracts handcrafted features and performs initial classification. This stage acts as a coarse-grained filter that can handle computationally inexpensive features.

**4.3.4 Stage 2: Preprocessing for mobile net.** The pipeline's second phase is dedicated to deep learning-based classification using MobileNet, a convolutional neural network renowned for its strong feature extraction and pattern recognition capabilities [50]. Unlike the RF model, MobileNet performs better with high-resolution images, emphasizing contrast and structural details. To optimize these inputs, we utilize a specialized enhancement-driven preprocessing pipeline, implemented via the `enhance_image()` function. This pipeline consists of the following steps:

 

1. **Background Removal:**

   To isolate the region of interest (ROI), we perform background removal using alpha matting [51]. Let $I_{input}(x, y)$ represent the original input image, and $I_{background}(x, y)$ denote the estimated background. The foreground image $I_{foreground}(x, y)$, containing the ROI, is obtained as:

$$I_{foreground}(x, y) = I_{input}(x, y) - \alpha(x, y) \cdot I_{background}(x, y), \tag{3}$$

   where $\alpha(x, y)$ is the alpha channel that controls the transparency of each pixel in the background.

2. **Contrast Limited Adaptive Histogram Equalization (CLAHE):**

   To enhance local contrast and improve the visibility of finer details, we apply CLAHE to the Value (V) channel of the HSV color space. CLAHE prevents noise over-enhancement by applying contrast clipping [52]. Let $V(i)$ represent the intensity histogram, then CLAHE transforms it as:

$$V'(i) = \min\left(\frac{V(i)}{clip\_limit}, 1\right), \tag{4}$$

   where clip_limit determines the upper threshold for histogram clipping. This adaptive equalization enhances underexposed and overexposed regions uniformly.

3. **Gamma Correction:**

   Gamma correction adjusts pixel intensities to enhance non-linear brightness, making subtle patterns more distinguishable [53]. The gamma-transformed pixel intensity $I_{gamma}(x, y)$ is computed as:

$$I_{gamma}(x, y) = 255 \cdot \left(\frac{I_{input}(x, y)}{255}\right)^{\gamma}, \tag{5}$$

   where $\gamma$ is set to 1.5 in this work. This boosts darker regions, enhancing contrast without amplifying noise excessively.

4. **Color Space Enhancement:**

   Following CLAHE and gamma correction, the processed image is converted back to RGB space to retain color-based information, which is crucial for CNN-based feature extraction [54]. The enhanced RGB image $I_{enhanced}$ preserves structural patterns and textural details, facilitating DenseNet's ability to learn high-level features.

**4.3.5 Hybrid decision model.** The overall architecture can be mathematically represented as a hybrid function $f_{Hybrid}(I)$, which makes a classification decision based on the outputs of both the Logistic Regression and MobileNet models:

$$y = f_{Hybrid}(I) = \begin{cases} f_{LR}(I_{norm}), & \text{if Stage 1 (Traditional Preprocessing),} \\ f_{MobileNet}(I_{enhanced}), & \text{if Stage 2 (Enhancement-Based Preprocessing).} \end{cases} \tag{6}$$

Here, $f_{LR}$ and $f_{MobileNet}$ denote the classification functions of the Logistic Regression and MobileNet models, respectively. The preprocessing steps are crucial in ensuring that $I_{norm}$ and $I_{enhanced}$ provide the appropriate inputs for each classifier.

Fig 16 shows the classification workflow of the designed decision-making process.

The proposed hybrid pipeline, which combines traditional feature extraction with deep learning, offers a reliable and efficient approach to disease detection. Its strength lies in balancing accuracy with computational efficiency, allowing even resource-limited devices to detect diseases in real time. The model adapts to various conditions and performs well on

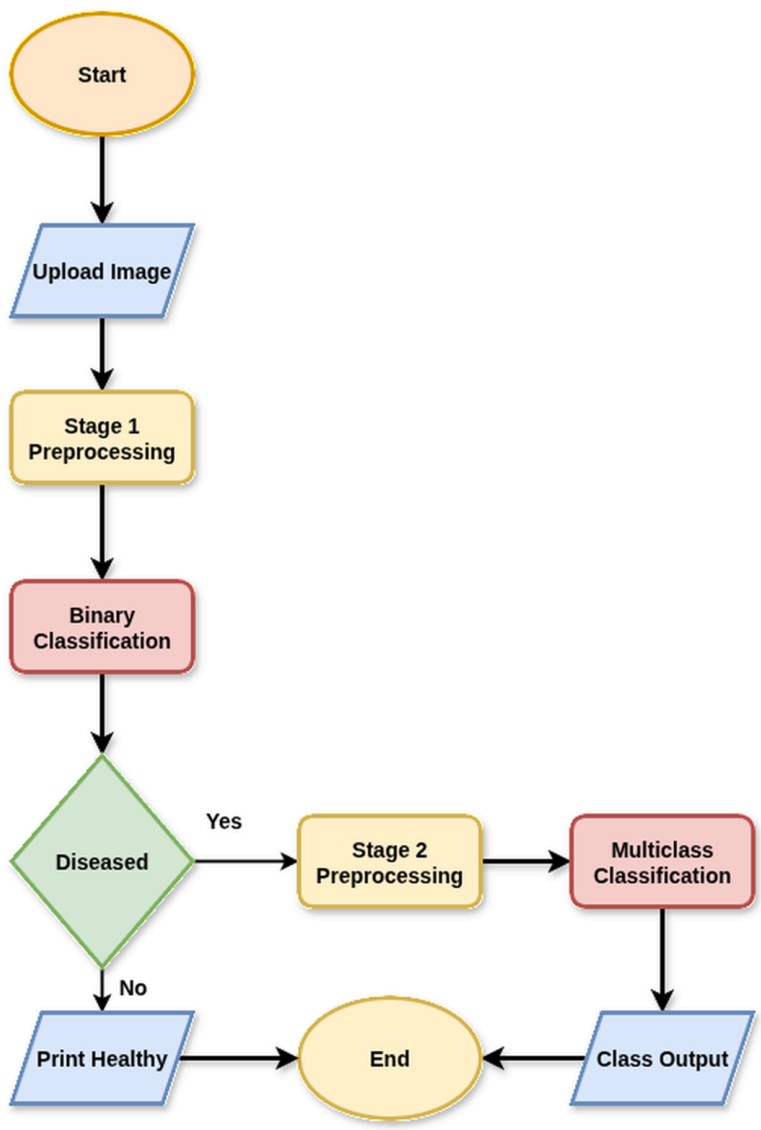

**Fig 16**. **Hybrid model workflow diagram.**

complex agricultural images with high variability by applying customized preprocessing steps at different stages. Beyond improving disease detection, this method offers a scalable and cost-effective solution that supports sustainable farming practices. Early detection is crucial in reducing resource waste and preventing crop losses, contributing to more efficient and environmentally friendly agricultural practices.

### 4.4 Preprocessing rational

The average processing time per loop for a single image was measured to evaluate the efficiency of the two prepro-cessing layers in the image enhancement and feature extraction pipeline. Additionally, the time difference and speed ratio between the layers were calculated. Table 8 presents a comparative analysis, outlining the time taken per loop, the number of iterations, the time difference, and the speed ratio.

**Table 8**. Comparative analysis of preprocessing time and iterations.

| Preprocessing Layer | Time per Loop | Number of Iterations |
|---|---|---|
| Stage 1 | 2.24 ms ± 54.6 $\mu$s | 100 loops |
| Stage 2 | 792 ms ± 164 ms | 100 loops |

The table shows that Layer 1 processes each loop in an average of 2.24 milliseconds (ms), while Layer 2 takes significantly longer, at 792 ms per loop. This results in a time difference of about 789.76 ms, making Layer 1 roughly 353 times faster than Layer 2. The difference in speed is due to the nature of their tasks. Layer 1 handles basic preprocessing steps, such as resizing, normalization, and pixel intensity adjustments, which require minimal computation. In contrast, Layer 2 employs more complex feature extraction techniques, including Local Binary Patterns (LBP), Hu Moments, and Histogram of Oriented Gradients (HOG), resulting in higher processing times.

The evaluation of the two-layer preprocessing strategy highlights the trade-off between processing speed and feature richness. While Layer 1 is much faster, Layer 2 captures more detailed and meaningful features that enhance disease classification accuracy. The hybrid model leverages the benefits of both, combining the speed of Layer 1 with the in-depth feature extraction capabilities of Layer 2. This integration optimizes the preprocessing pipeline, making it well-suited for real-time agricultural applications where both performance and efficiency are critical.

## 5 Result analysis and discussion

The following analysis presents a detailed evaluation of the proposed hybrid model, focusing on its performance, computational efficiency, and scalability in various deployment scenarios. The results are compared against a benchmark single-stage MobileNet model to validate the advantages of the two-stage hybrid architecture.

### 5.1 Performance of the hybrid model

The performance of the proposed hybrid model was first assessed against a single MobileNet model using 5-fold cross-validation on a controlled test dataset. As summarized in Table 9 and visualized in Fig 17, the hybrid model achieves a mean accuracy of 91.57%, while the single MobileNet model reaches 94.63%. Although the single MobileNet model exhibits a slightly higher accuracy, the hybrid model's performance is notable for its significant gain in computational efficiency. For this, 600 images are used from the test set, with each class comprising 75 images.

The bar chart in Fig 17 visually confirms the code trade-off of the hybrid approach: a slight reduction in accuracy for a substantial gain in speed. The hybrid model's average inference time is 34.97 seconds, representing a significant reduction compared to the 64.57 seconds required by the single MobileNet model. The paired t-test results in Table 10 confirm that both the observed difference in accuracy (p = 0.037) and inference time (p = 0.00006) are statistically significant. This finding provides a rigorous, data-driven validation for the model design, confirming that the trade-off is well-justified by the substantial gain in efficiency.

**Table 9**. 5-Fold cross-validation results on test dataset (controlled use case GPU).

| Model | Fold 1 | Fold 2 | Fold 3 | Fold 4 | Fold 5 |
|---|---|---|---|---|---|
| MobileNet Accuracy (%) | 93.86 | 97.71 | 94.29 | 94.57 | 92.71 |
| MobileNet Time (s) | 72.12 | 65.12 | 62.12 | 62.39 | 61.09 |
| Hybrid Model Accuracy (%) | 92.13 | 90.79 | 91.53 | 92.23 | 91.17 |
| Hybrid Model Time (s) | 37.85 | 32.17 | 34.50 | 35.92 | 34.41 |

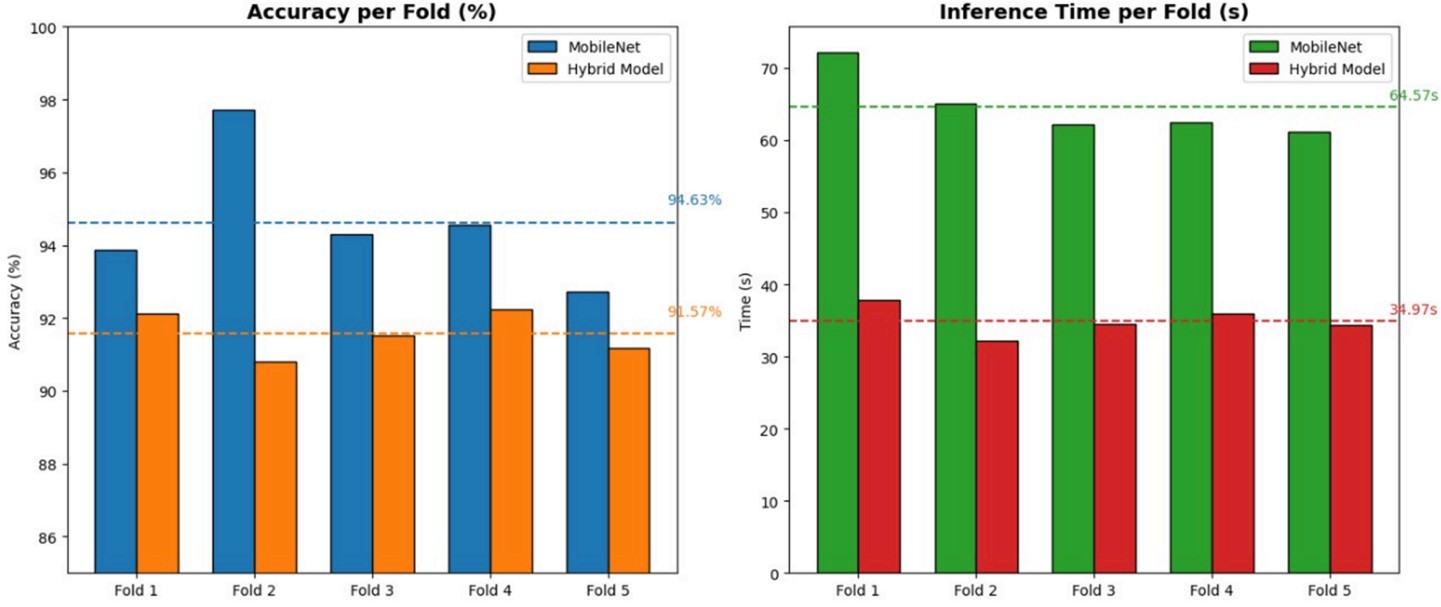

**Fig 17. Model performance (controlled use-case).**

**Table 10. Paired t-test results comparing MobileNet and hybrid model.**

| Metric | t-Statistic | p-Value |
|---|---|---|
| Accuracy (MobileNet vs Hybrid) | 3.09 | 0.037 |
| Inference Time (MobileNet vs Hybrid) | 17.7 | 0.00006 |

## 5.2 Scalability and resource consumption

To evaluate the practical viability of the proposed approach, a "pseudo-random use case" was simulated using 1,227 images selected at random, with approximately 40% of the photos being healthy. The models were then deployed on three different hardware platforms: a Desktop and two laptops mentioned above, on CPU, and using the ONNX runtime.

The results, presented in Table 11, highlight the remarkable scalability and efficiency of the hybrid model across varying computational environments. To create a neutral experiment environment, turbo boost and C-state were disabled across all devices. The desktop processor was locked at 65 watts TDP. Laptop one and laptop two processors were locked at 28 watts and 15 watts, respectively. High-performance mode was selected in the Operating system's power manager.

This test was explicitly designed to isolate and quantify computational performance, using a random subset of the data to measure real-world inference speed while validating the system's fundamental accuracy.

**Table 11. Performance comparison for random use case across devices.**

| Platform | Model | Accuracy (%) | Time (s) |
|---|---|---|---|
| PC | MobileNet | 78.46 | 1073.77 |
| PC | Hybrid | 91.53 | 478.61 |
| Laptop one | MobileNet | 78.46 | 1302.56 |
| Laptop one | Hybrid | 91.53 | 515.14 |
| Laptop two | MobileNet | 78.46 | 4548.35 |
| Laptop two | Hybrid | 91.53 | 1010.13 |

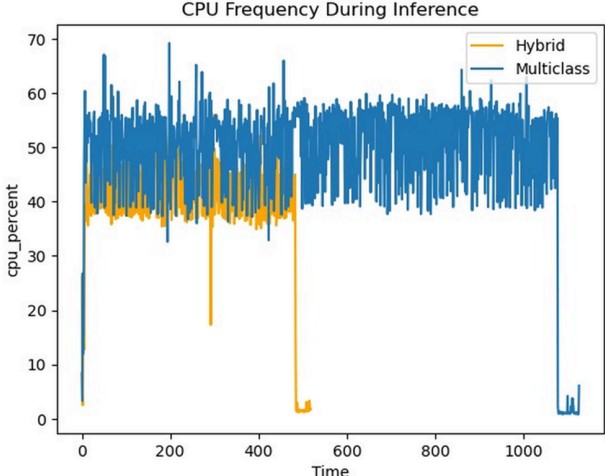

Across all hardware platforms, the hybrid model demonstrated a marked efficiency advantage. The per-image inference time on the PC was reduced from 0.874 seconds (MobileNet) to just 0.3896 seconds (Hybrid Model). This translated to a 55.4% reduction on the PC. The efficiency gains were even more significant on the entry-level laptops, with the hybrid model achieving a 60.4% reduction on Laptop One and an impressive 77.6% reduction on Laptop Two, as detailed in Table 12.

The computational efficiency of the hybrid model is further substantiated by a detailed analysis of CPU resource utilization during the inference process, providing crucial evidence of the model's sustainable design.

The CPU Frequency During Inference plot in Fig 18 clearly demonstrated the efficiency gains. The hybrid model (yellow line) maintains a significantly lower and more consistent CPU frequency compared to the multiclass model (blue line) throughout the inference period. This reduced demand on the processor directly translates to lower power consumption and less heat generation.

This decreased load is further confirmed by the CPU Usage Over Time chart Fig 19. The hybrid model exhibits a lower overall percentage of CPU usage and far less fluctuation. By minimizing the computational demands on the CPU, the hybrid architecture enables robust deployment on commodity and low-power hardware, reinforcing the paper's core argument that high-performance AI for agriculture can be both accessible and environmentally responsible. Additionally, we evaluated the performance using various configurations of leaf types, with a specific focus on healthy leaves. As shown in Table 13, under different conditions, our proposed architecture consistently outperformed others, demonstrating significantly faster performance on lower-end devices.

## 5.3 Observation in isolation on test set

To understand the performance improvements of the hybrid model, a detailed study was conducted to examine the model's behavior and the impact of preprocessing. A Random sample of 58 images from the test set was selected for this

**Table 12. Per-image inference time and time reduction for hybrid model vs MobileNet.**

| Platform | MobileNet (s) | Hybrid (s) | Reduction (%) |
|---|---|---|---|
| PC | 0.874 | 0.3896 | 55.4% |
| Laptop One | 1.06 | 0.4198 | 60.4% |
| Laptop Two | 3.71 | 0.83 | 77.6% |

**Fig 18. Confusion matrix Mobilenet model (test data).**

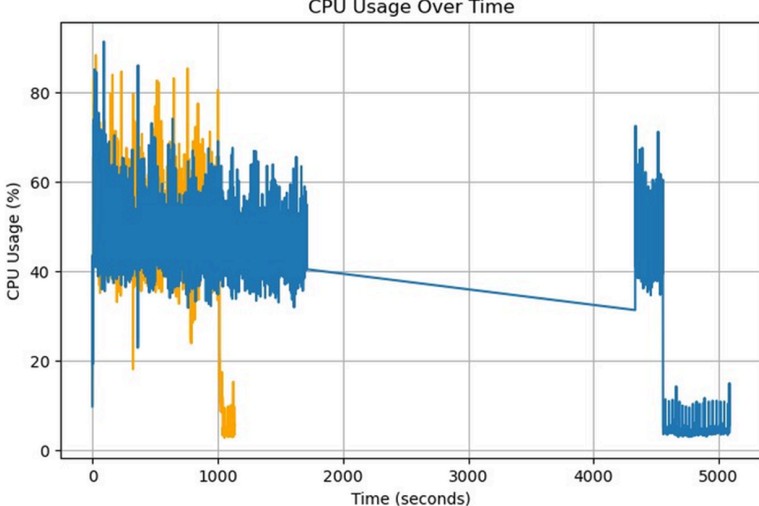

**Fig 19**. **Confusion matrix Mobilenet model (Test data).**

**Table 13**. **Performance of different devices at varying levels.**

| Device | 50% Healthy | 60% Healthy | 70% Healthy |
|---|---|---|---|
| PC | 67.0 | 70.0 | 74.8 |
| Laptop 1 | 66.5 | 72.0 | 77.3 |
| Laptop 2 | 79.6 | 80.4 | 81.4 |

purpose. The tests were performed on a desktop CPU, with inference time recorded only for the deep learning models, not the entire pipeline and preprocessing.

Table 14 illustrates that the hybrid model demonstrates a notable efficiency advantage, completing the task 47.02% faster than the MobileNet model during inference. While the MobileNet model achieves a slightly higher accuracy of 93.10% compared to the hybrid model's 91.38%, this minor difference is a key trade-off for the substantial gain in speed.

A more revealing analysis was conducted by evaluating the model's performance without the preprocessing steps (Rembg and CLAH). The results in Table 15 highlight a significant finding: the MobileNet model's accuracy plummets to

**Table 14**. **Performance comparison between hybrid model and MobileNet (test set).**

| Model | Accuracy (%) | Inference Time (ms) |
|---|---|---|
| Hybrid Model | 91.38 | 270.88 |
| MobileNet | 93.10 | 520.62 |
| Hybrid vs MobileNet | 2.62% | 47.02% faster |

**Table 15**. **Performance comparison between hybrid model and MobileNet (without Rembg and CLAHE).**

| Model | Accuracy (%) | Inference Time (ms) |
|---|---|---|
| Hybrid Model | 91.53 | 270.88 |
| MobileNet | 56.46 | 380.80 |
| Hybrid vs MobileNet | 35.07 | 28.14% faster |

56.46% without these techniques. In contrast, the hybrid model maintains a high accuracy of 91.53% with a 28.14% faster inferencing time while these preprocessing steps are active, demonstrating remarkable resilience to their effects. This indicates that the hybrid model's two-stage architecture is inherently more robust.

This disparity in performance is visually confirmed by the confusion matrices for both models as detailed in Figs 20 and 21.

The difference in performance is clearly seen in the confusion matrices. The MobileNet matrix in Fig 20 reveals major misclassifications, particularly struggling to correctly identify the Healthy class (with only nine correct classifications, and three misclassified as Powdery Mildew). Furthermore, Gall Midge identification was severely compromised, with three instances misclassified as Dieback. In sharp contrast, the Hybrid Model matrix in Fig 21 maintains a very high number of correct predictions along the main diagonal (e.g., 11 for healthy, 8 for Die Back). This visual proof confirms the hybrid model's superior robustness, demonstrating its reliable classification ability even when crucial external preprocessing is removed. All the codes are available online to support reproducibility [55].

## 5.4 Broader implications

This research addresses a critical gap in current agricultural AI: the need for solutions that are not only accurate but also accessible, sustainable, and scalable. While many existing studies achieve high accuracy using single-stage deep

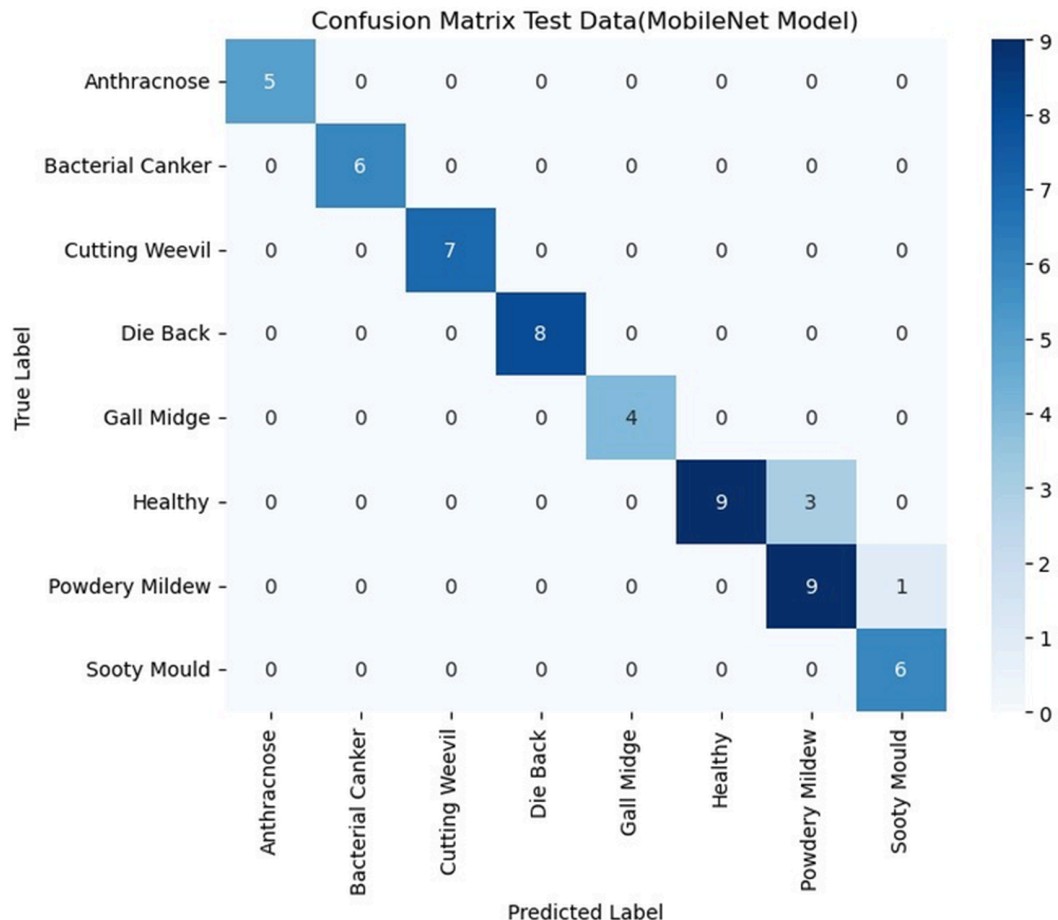

**Fig 20**. Confusion matrix Mobilenet model (test data).

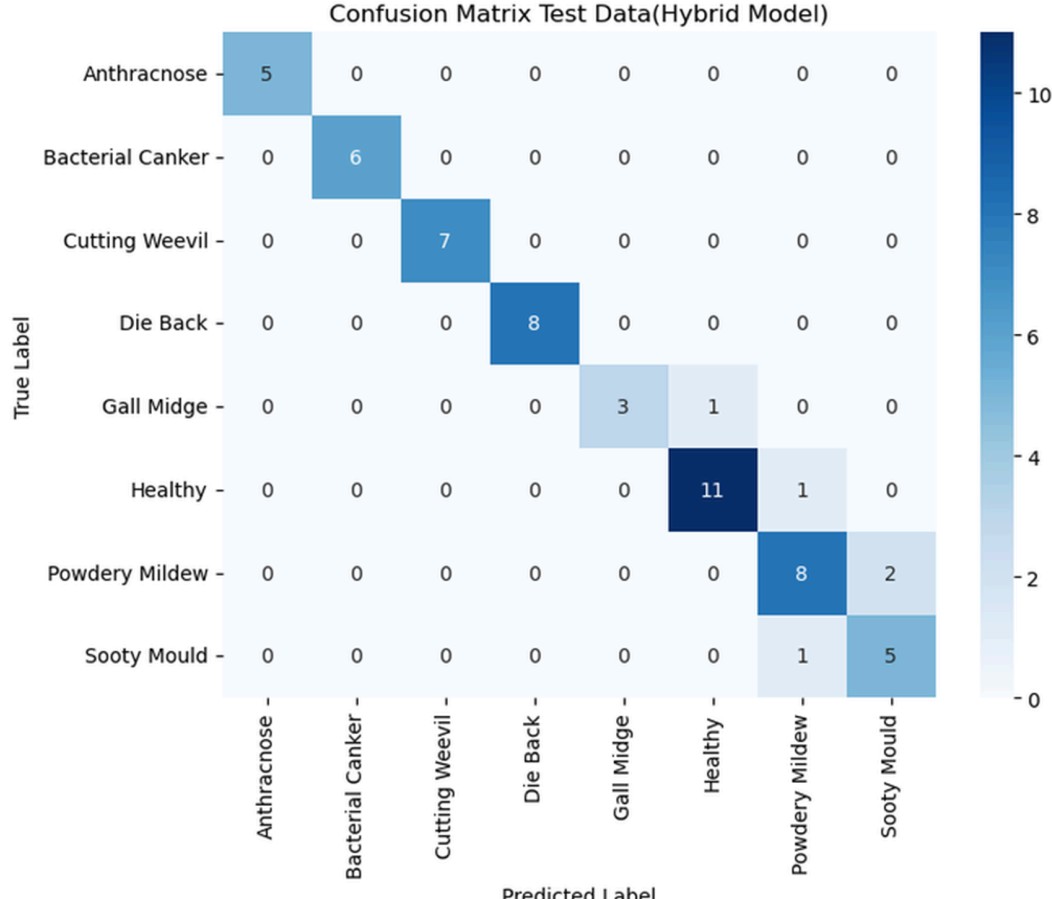

**Fig 21**. Confusion matrix hybrid model (test data).

learning models, they often rely on computationally expensive infrastructure that is inaccessible to farmers in low-resource settings. The core significance of this work lies in demonstrating that a hybrid, dual-stage architecture can overcome this limitation without incurring a significant performance sacrifice.

The substantial reduction in inference time and CPU load, particularly on commodity hardware, positions this work as a significant step toward developing practical AI solutions for the agricultural sector.

## 6 Sustainable Development Goals (SDGs) and High-Performance Computing (HPC)

This study aligns with the Sustainable Development Goals (SDGs), particularly SDG 2 (Zero Hunger) and SDG 12 (Responsible Consumption and Production). The proposed hybrid model for leaf disease detection plays a significant role in promoting sustainable agriculture by offering efficient and accurate methods for the early detection of diseased plants. This capability significantly enhances agricultural productivity while minimizing environmental impacts.

### 6.1 SDG 2 (Zero Hunger)

The hybrid model supports SDG 2, which enables faster and more efficient disease detection. It helps farmers make timely and informed decisions about crop health and management. Identifying diseases early is essential for minimizing

crop losses, ensuring food security, and preventing widespread damage [56]. With accurate and reliable detection tools, farmers can take action before an outbreak spreads, ultimately leading to higher yields and a more stable food supply.

## 6.2 SDG 12 (Responsible consumption and production)

The hybrid model aligns with SDG 12 by promoting responsible resource consumption. Its reduced inference time and smaller model size help lower energy use, making it more efficient and environmentally friendly. Additionally, its ability to detect plant diseases early minimizes the need for excessive pesticide use, supporting more sustainable farming practices and reducing the harmful effects of chemical overuse on the environment [57].

## 7 Explainable AI and iterative enhancement for efficient disease detection

The most important aspect of this methodology is the use of iterative enhancement and Explainable AI (XAI) techniques. These techniques enhance model performance and offer valuable insights into informed decision-making. The iterative enhancement process gradually refines the model by learning from errors and adjusting at each stage. The model becomes more accurate and efficient over time by continually learning from misclassified instances, utilizing preprocessing and feature selection techniques.

This method helps identify the key features or areas within an image that have the most significant impact on classification decisions. Adding explainability to the process clarifies how the model arrives at its predictions and which elements play the most critical role. This transparency is crucial for fine-tuning the model, as it identifies inefficiencies or biases, improving accuracy and dependability.

The Iterative Enhancement method focuses on improving extracted features and how the model learns from them. Initially, the model depends on traditional preprocessing methods. Over time, through iterative improvements, it begins to incorporate more advanced techniques, like deep feature learning. This helps the model better understand the complex patterns in diseased leaves. With each iteration, the model refines its ability to recognize different disease patterns, gradually becoming more efficient and accurate over time.

## 8 High-Performance Computing (HPC)

The proposed hybrid model integrates high-performance computing (HPC) principles, ensuring computational efficiency through iterative enhancements and explainability.

The reduced inference time and smaller model. size improves computational performance, making the hybrid model more suitable for real- time deployment in field environments, particularly in resource-constrained areas. This is particularly useful in developing countries where access to high-end computational resources is limited. The optimized inference time and reduced model size allow the model to run efficiently on devices with lower computing power, ensuring accessibility and effectiveness across various agricultural contexts.

Moreover, integrating the ONNX format for MobileNet enhances the model's compatibility across different hardware platforms, making deployment more flexible. This adaptability is crucial for scaling its use across various agricultural settings.

## 9 Environmental sustainability and energy-efficient computing

The hybrid model is designed to be both efficient and eco-friendly. It reduces energy consumption during inference by optimizing its deep learning and machine learning components—Random Forest and MobileNet in ONNX format. This is especially important in large-scale farming, where high energy use can increase costs.

To ensure sustainability, the model undergoes continuous refinement to enhance its accuracy while using fewer resources. Its ability to deliver precise results with minimal energy makes it a practical choice for real-time agricultural

applications. This approach supports the broader goal of making AI-driven solutions more energy-efficient and environmentally friendly.

## 10 Deployment

The results of this study have strong potential for real-world application in agriculture, extending beyond controlled laboratory environments. The images used in the dataset were collected from different mango trees under natural conditions without any special setup. This reflects how farmers typically take pictures using phones or other personal devices, often in unpredictable environments. The models were tested on regular personal computers rather than servers or expensive machines to make the research more practical. This makes the approach more realistic and accessible for use in actual farming conditions. One of the key highlights of the study is the models' ability to make predictions quickly. MobileNet consistently delivered the fastest performance among the three models tested, as shown in Table 2. Speed is especially crucial in agriculture, where the rapid detection of leaf diseases can make a significant difference. MobileNet's speed and efficiency make it a strong candidate for real-time applications in the field.

To bridge the divide between research and practical application, we have developed an application, as shown in Fig 22. It is easy for users to navigate. This application leverages the capabilities of our trained algorithm, allowing users to capture photographs of mango leaves effortlessly. It can be ported to other devices, such as personal mobile phones and edge devices. The application then analyzes the image and instantly delivers information regarding the presence of diseases. By leveraging the efficient inference time of the MobileNet model, users can quickly and accurately identify plant diseases. This enables them to make well-informed decisions regarding plant health management.

## 11 Conclusion

The proposed two-stage hybrid pipeline provides a practical solution to several key challenges in plant disease detection, particularly in real-world and resource-constrained agricultural environments. Combining traditional machine learning with deep learning and customized preprocessing steps, the system achieves high classification accuracy while maintaining fast and efficient inference. Additionally, MobileNet was converted to ONNX format, forming the core of the deep learning stage, while a lightweight Random Forest classifier filters healthy leaves early on, minimizing computational load. To ensure the model is not just accurate but also transparent, Grad-CAM highlights the regions of the leaf that most influence predictions. These visual explanations helped to understand the decision-making process, making the system more trustworthy and easier to refine over time. During the scalability test, the pipeline operates effectively on mango leaf images, demonstrating strong performance, scalability, and compatibility with edge devices at different experimental configurations. The model achieved an inferencing speed 77%-80% faster than the baseline, making it suitable for real-world use cases. Alongside its technical strengths, the approach supports global sustainability efforts, including SDG 2 (Zero Hunger) and SDG 12 (Responsible Consumption and Production), enabling faster and more efficient disease detection. Its modular design enables easy adaptation for detecting diseases in other crops and plants, expanding its relevance for scalable and sustainable agricultural systems. Additionally, this experiment focuses on commodity hardware for deployment, namely, a personal laptop. This removes the barrier for adaptation in low-income areas as well as reduces the cost and time of disease detection.

While the pipeline demonstrates promising results on mango leaves, its adaptability can be further tested on other crops with similar disease profiles in the future. Testing on datasets from different countries could further strengthen generalizability; however, since mango production is concentrated primarily in the Indian subcontinent, the current dataset provides a strong foundation for evaluation. Moreover, the present implementation processes images sequentially, which constrains throughput when working with large datasets or real-time streams. Future work will therefore focus on incorporating parallel processing strategies to enhance performance and scalability under higher data loads.

# Hybrid Disease Detection System

Upload Plant Image

Drag and drop file here
Limit 200MB per file • JPG, JPEG, PNG

Browse files

IMG_20211106_120700 (Custom).jpg  33.3KB  ✕

Initial Screening Result: Diseased

## Detailed Analysis

The `use_column_width` parameter has been deprecated and will be removed in a future release. Please utilize the `use_container_width` parameter instead.

The `use_column_width` parameter has been deprecated and will be removed in a future release. Please utilize the `use_container_width` parameter instead.

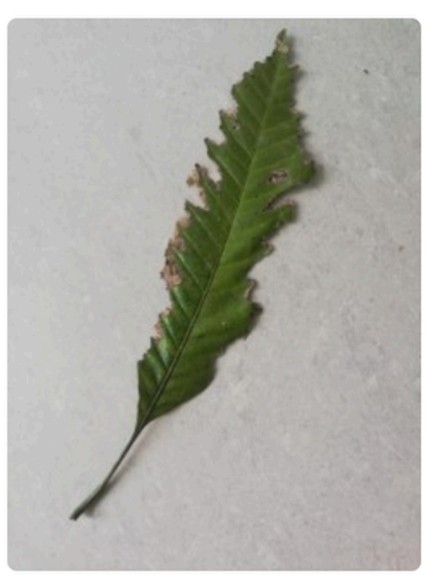
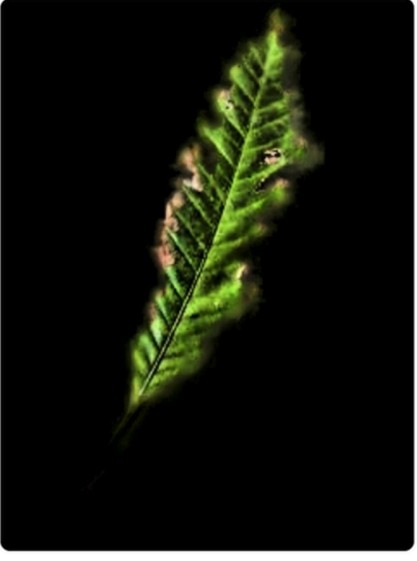

Original Image

Enhanced Image

Detailed Diagnosis: Bacterial Canker

Recommended Actions: Consult with agricultural expert for targeted treatment.

**Fig 22**. **Prototype interface of the application used for real-time disease detection in mango leaves.**

## Author contributions

**Conceptualization:** Md Abdullah Al Kafi, Sumit Kumar Banshal.

**Data curation:** Raka Moni.

**Formal analysis:** Md Abdullah Al Kafi, Sumit Kumar Banshal.

**Funding acquisition:** Aulia Luqman Aziz, Mohammed Aljuaid, Rohit Bansal.

**Investigation:** Raka Moni, Mohammed Aljuaid.

**Methodology:** Md Abdullah Al Kafi, Raka Moni.

**Project administration:** Sumit Kumar Banshal, Mohammed Aljuaid.

**Resources:** Md Abdullah Al Kafi, Aulia Luqman Aziz, Rohit Bansal.

**Software:** Md Abdullah Al Kafi, Raka Moni, Rohit Bansal.

**Supervision:** Sumit Kumar Banshal.

**Validation:** Md Abdullah Al Kafi, Aulia Luqman Aziz.

**Writing – original draft:** Md Abdullah Al Kafi, Raka Moni.

**Writing – review & editing:** Sumit Kumar Banshal.

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
