## [Decision Letter · Decision Letter 0]

8 Jun 2025

PONE-D-25-23549LeafAI: Interpretable Plant Disease Detection for Edge ComputingPLOS ONE

Dear Dr. Banshal,

Thank you for submitting your manuscript to PLOS ONE. After careful consideration, we feel that it has merit but does not fully meet PLOS ONE’s publication criteria as it currently stands. Therefore, we invite you to submit a revised version of the manuscript that addresses the points raised during the review process.

We look forward to receiving your revised manuscript.

Kind regards,

Asadullah Shaikh, Ph.D.

Academic Editor

PLOS ONE

Journal Requirements:

“We would like to extend our appreciation to King Saud University for funding this work through the Researcher Supporting Project (RSP2025R481), King Saud University, Riyadh, Saudi Arabia”

“We would like to extend our appreciation to King Saud University for funding this work through the Researcher Supporting Project (RSP2025R481), King Saud University, Riyadh, Saudi Arabia”

“The author(s) received no specific funding for this work”

7. We note you have included a table to which you do not refer in the text of your manuscript. Please ensure that you refer to Table 5 in your text; if accepted, production will need this reference to link the reader to the Table.

Reviewers' comments:

Reviewer's Responses to Questions

**Comments to the Author**

1. Is the manuscript technically sound, and do the data support the conclusions?

Reviewer #1: Yes

Reviewer #2: No

Reviewer #3: Partly

Reviewer #4: Yes

Reviewer #5: Yes

2. Has the statistical analysis been performed appropriately and rigorously?

Reviewer #1: Yes

Reviewer #2: No

Reviewer #3: Yes

Reviewer #4: Yes

Reviewer #5: Yes

3. Have the authors made all data underlying the findings in their manuscript fully available?

Reviewer #1: No

Reviewer #2: No

Reviewer #3: Yes

Reviewer #4: Yes

Reviewer #5: Yes

4. Is the manuscript presented in an intelligible fashion and written in standard English?

Reviewer #1: Yes

Reviewer #2: No

Reviewer #3: No

Reviewer #4: Yes

Reviewer #5: Yes

5. Review Comments to the Author

Reviewer #1: Strengths of the Study

Innovative Hybrid Architecture

The two-stage system combining Random Forest (for healthy vs. diseased leaf filtering) and DenseNet (for disease classification) is well-motivated, addressing real-world class imbalance and efficiency.

Emphasis on Explainability

Use of Grad-CAM enhances interpretability, helping visualize what features the model relies on. This is critical in building trust for AI adoption in agriculture.

Deployment Focus

Practical attention to edge computing and resource-limited environments, with ONNX optimization and real-time inference on low-power hardware, makes the research application-ready.

Scalability and Sustainability

The study addresses SDGs (especially SDG 2 and SDG 12), focusing on low-energy AI, real-time detection, and potential mobile deployment, aligning with sustainable and global agricultural goals.

Real-World Dataset

Use of field-collected mango leaf images under varied natural conditions increases the generalizability of the model compared to lab-curated datasets like PlantVillage.

Recommendations for Revision

1. Add a simplified schematic or flowchart of the hybrid model for readers to quickly understand the pipeline.

2. Edge computing involves low-powered device like Raspberry Pi. Laptops are more pwerful device to deploy deep learning models. To validate the practical applicability of the proposed model in real-world field conditions, it is strongly recommended to deploy and test the system on a true edge computing device such as a Raspberry Pi 4, NVIDIA Jetson Nano, or Google Coral Edge TPU. These platforms offer limited computational resources and power consumption profiles that more accurately reflect the constraints of agricultural edge environments.

The trained DenseNet model can be optimized through quantization (e.g., using TensorFlow Lite or ONNX format) and deployed on the selected edge hardware. Real-time inference tests should be conducted using a live camera feed, capturing leaf images under natural lighting. Key performance metrics such as inference latency, memory usage, CPU utilization, and energy efficiency should be recorded and compared to the laptop-based benchmarks. This would ensure the system meets the operational requirements for real-time, low-power plant disease detection in the field.

Reviewer #2: SUMMARY

This manuscript proposes a hybrid AI approach combining Random Forest and deep learning for plant disease detection. While addressing a relevant agricultural problem, the work contains fundamental methodological flaws that prevent publication.

MAJOR ISSUES

1. Experimental Design Deficiencies

No independent test set validation

Missing cross-validation or statistical significance testing

Contradictory statements about class imbalance (abstract vs. Section 3.4)

Single-run experiments without proper controls

2. Non-compliance with PLOS ONE Requirements

Complete absence of data availability statement

No access to datasets, code, or trained models

Insufficient detail for reproducibility

3. Inadequate Evaluation

Only accuracy reported; missing precision, recall, F1-score

No confusion matrices or error analysis

Limited baseline comparisons (only DenseNet121)

Unsubstantiated performance claims ("70× faster")

4. Statistical Analysis Deficiencies

No confidence intervals or uncertainty quantification

Missing statistical tests for model comparisons

Insufficient sample size documentation

MINOR ISSUES

Language quality issues throughout

Inconsistent terminology usage

Formatting errors in citations and figures

DECISION RATIONALE

The experimental validation is fundamentally flawed, making all performance claims unreliable. The absence of proper statistical analysis and failure to meet data sharing requirements are blocking issues that cannot be addressed through revision alone.

Reviewer #3: This paper presents an iterative, hybrid AI approach. The hybrid system operates in two stages: first, a lightweight random forest classifier performs binary classification to quickly separate and exclude healthy leaves; then, a deep learning model classifies specific diseases into smaller groups of diseased leaves. The system combines traditional machine learning with deep learning, combined with customized preprocessing steps, to achieve high classification accuracy while maintaining fast and efficient reasoning capabilities.

However, the shortcomings of this paper include:

1. The "edge computing" mentioned in the title is not reflected in the abstract.

2. The "food safety" in the "keywords" is also not reflected in this paper.

3. The description of the 8 types of diseased leaves in the "Data Description" section is too lengthy, and it is recommended to be concise and summarized.

4. The "Introduction" section of this paper needs to strengthen the comparative analysis with existing research and clearly list the core innovations and contributions of the proposed method.

5. It is recommended to fully display the 7 diseased leaf heat maps used in this article in 'Fig 2' to ensure the consistency of data presentation in the full text.

6. The "edge computing" in the title is not discussed enough in the article. It is recommended to add relevant content to highlight the contribution of "edge computing" in this article.

7. The number of charts in this article is small and the expressiveness is limited. It is recommended to add high-quality charts (such as process diagrams, data visualization, etc.) to enhance the intuitiveness and persuasiveness of the discussion.

8. Please further polish the language without colloquial words.

9. It is recommended to add more model detection accuracy and reasoning speed comparisons, and use charts to show the superiority of the method used in this article in detection accuracy and reasoning speed.

Reviewer #4: 1. The research content of the manuscript significantly overlaps with previous studies, and the level of innovation is insufficient. In particular, the proposed method does not demonstrate clear improvements over existing approaches.

2. The Methods section of the manuscript is vague, lacking adequate mathematical definitions and symbol explanations.

3. The authors need to explain the necessity of using edge computing techniques in leaf disease detection.

4. The analysis lacks depth. In particular, when performance differences are observed, the manuscript fails to explain the underlying reasons or provide theoretical justification.

5. The manuscript contains grammatical errors, and some paragraphs are logically confusing.

Reviewer #5: 1. Details about the dataset and evaluation metrics in the abstract would be appreciated.

2. Paper mentions up to 70x faster inference, but including quantitative performance metrics and comparisons with existing models would be valuable.

3. Clarify how adaptable the framework is to different crops beyond mango leaves, possibly with a discussion.

4. A more detailed description of the hybrid model architecture, the training process, and the interpretability methods would be beneficial for reproducibility.

6. PLOS authors have the option to publish the peer review history of their article (what does this mean?). If published, this will include your full peer review and any attached files.

Reviewer #1: No

Reviewer #2: No

Reviewer #3: No

Reviewer #4: No

Reviewer #5: No

---

## [Author Response · Author response to Decision Letter 1]

25 Jul 2025

Response to Reviewers

Journal Requirements:

Response: We regret that the template was not matched earlier; however, we have adhered to the styling and format options as mentioned in the provided links.

2. Please note that PLOS ONE has specific guidelines on code sharing for submissions in which author-generated code underpins the findings in the manuscript. In these cases, we expect all author-generated code to be made available without restrictions upon publication of the work. Please review our guidelines at https://journals.plos.org/plosone/s/materials-and-software-sharing#loc-sharing-code and ensure that your code is shared in a way that follows best practice and facilitates reproducibilitxy and reuse.

Response: We have added the direct link to the code in the methodology section, and we have mentioned the same in our data availability statement as per the policy.

Response: We have added the funding information section at the end.

“We would like to extend our appreciation to King Saud University for funding this work through the Researcher Supporting Project (RSP2025R481), King Saud University, Riyadh, Saudi Arabia”

Response: The funder has took role in preparing the finalized draft during the first draft. Also during the revision, funder helped in revising the manuscript effectively.

“We would like to extend our appreciation to King Saud University for funding this work through the Researcher Supporting Project (RSP2025R481), King Saud University, Riyadh, Saudi Arabia”

“The author(s) received no specific funding for this work”

Response: We have added the funding information section at the end.

Response: We have added the direct link to the data in the methodology section, and we have mentioned the same in our data availability statement as per the policy.

7. We note you have included a table to which you do not refer in the text of your manuscript. Please ensure that you refer to Table 5 in your text; if accepted, production will need this reference to link the reader to the Table.

Response: Thank you for highlighting this issue, it has been addressed in the revised manuscript.

Recommendations for Revision

1. Add a simplified schematic or flowchart of the hybrid model for readers to quickly understand the pipeline.

Response: In the methodology section, a general workflow is added. In Hybrid decision model hybrid workflow model diagram is added.

2. Edge computing involves low-powered device like Raspberry Pi. Laptops are more pwerful device to deploy deep learning models. To validate the practical applicability of the proposed model in real-world field conditions, it is strongly recommended to deploy and test the system on a true edge computing device such as a Raspberry Pi 4, NVIDIA Jetson Nano, or Google Coral Edge TPU. These platforms offer limited computational resources and power consumption profiles that more accurately reflect the constraints of agricultural edge environments.

The trained DenseNet model can be optimized through quantization (e.g., using TensorFlow Lite or ONNX format) and deployed on the selected edge hardware. Real-time inference tests should be conducted using a live camera feed, capturing leaf images under natural lighting. Key performance metrics such as inference latency, memory usage, CPU utilization, and energy efficiency should be recorded and compared to the laptop-based benchmarks. This would ensure the system meets the operational requirements for real-time, low-power plant disease detection in the field.

Response : Thank you for the valuable suggestion. We fully agree that testing on low-powered edge devices like the Raspberry Pi 4, NVIDIA Jetson Nano, or Google Coral Edge TPU provides essential insight into real-world deployment scenarios. However, the primary aim of our research is to develop a practical and sustainable solution that farmers in low-income regions can adopt, in line with the United Nations Sustainable Development Goals (SDGs).

To that end, we deliberately chose a general-purpose, commodity laptop as our deployment platform. While not as resource-constrained as typical edge boards, it balances accessibility, affordability, and computational capability, making it a feasible choice for real-world use without cloud infrastructure. Since the model runs entirely on the device without server support, we consider it a valid edge deployment scenario.

Additionally, using specialized hardware or deploying clusters to meet strict performance metrics, such as energy efficiency or latency under minimal resources, could increase cost and complexity. This runs counter to our goal of creating a scalable, low-cost solution.

In response to the reviewer’s point, we have revised the discussion section to include key performance metrics—inference latency, memory usage, and CPU utilization—measured on the selected laptop platform.

Major Issues

1. Experimental Design Deficiencies

No independent test set validation

Missing cross-validation or statistical significance testing

Contradictory statements about class imbalance (abstract vs. Section 3.4)

Single-run experiments without proper controls

Response :We appreciate the reviewer’s detailed feedback and the opportunity to clarify these points. In the revised manuscript, we have addressed each concern as follows:

1. Independent Test Set Validation:

We have now included a clearly defined and separate test set to evaluate the final model performance. This test set was not used during training or cross-validation and was employed exclusively for final evaluation to ensure unbiased performance reporting. Details of the test set usage are provided in Section 3.2 (Experimental Setup) and Section 4 (Result Analysis).

2. Cross-Validation:

We have implemented 5-fold cross-validation across all model comparisons, including MobileNet, ResNet50, and DenseNet121, as shown in Table 2 and Figure 3. Each fold was trained and validated independently to ensure robustness and generalizability of the results.

3. Statistical Significance Testing:

While the original version lacked statistical testing, we have now included standard deviation values for accuracy and inference time across folds (see Table 5).

4. Clarification on Class Imbalance:

Thank you for pointing out the inconsistency. We have revised the abstract and Section 3.4 to ensure alignment. The abstract now clearly states that the class imbalance is leveraged through a two-stage hybrid pipeline, where the Random Forest classifier filters out the majority class (healthy leaves), reducing the computational burden on the deep learning model. This is now consistently reflected throughout the manuscript.

5. Single-Run Experiments:

We acknowledge the importance of repeatability. All reported results are now averaged over multiple runs (cross-validation folds), and we have removed any single-run performance claims. This ensures that the reported metrics reflect consistent model behavior rather than isolated outcomes.

2. Non-compliance with PLOS ONE Requirements

Complete absence of data availability statement

No access to datasets, code, or trained models

Insufficient detail for reproducibility

Response: We thank the learned reviewer for highlighting these important compliance issues. We have taken the following steps to fully align the manuscript with PLOS ONE’s data sharing and reproducibility requirements:

1. Data Availability Statement:

We have now included a formal Data Availability Statement in the revised manuscript, placed immediately after the abstract, as per PLOS ONE guidelines. The statement reads:

1. Data Availability:

All relevant data are within the manuscript and its Supporting Information files. The MangoLeafBD dataset used in this study is publicly available at https://data.mendeley.com/datasets/hxsnvwty3r/1.

2. Access to Code and Trained Models:

To promote transparency and reproducibility, we have made the complete source code, trained model weights (for MobileNet and Random Forest), and preprocessing scripts publicly available on GitHub. The repository is accessible at:

3. GitHub - abkafi1234/Disease_Agnostic_Hybrid_classifier

This link has also been included in the manuscript under the “Data Availability” and “Deployment” sections.

4. Reproducibility Enhancements:

We have revised the Methodology section to include:

o Detailed descriptions of preprocessing steps (Sections 3.4.1 and 3.4.2), including equations and parameter values.

o Hardware specifications for both training and inference environments (Table 1).

o Clear explanation of the two-stage hybrid architecture and decision logic (Section 3.4.3).

o Cross-validation setup and test set separation (Section 3.2 and Section 4).

2. These additions ensure that the study can be independently replicated using the provided dataset and codebase.

We believe these revisions now bring the manuscript into full compliance with PLOS ONE’s policies on data availability and reproducibility. We appreciate the reviewer’s attention to these critical aspects of scientific transparency

3. Inadequate Evaluation

Only accuracy reported; missing precision, recall, F1-score

No confusion matrices or error analysis

Limited baseline comparisons (only DenseNet121)

Unsubstantiated performance claims ("70× faster")

Response: We appreciate the reviewer’s insightful feedback regarding the evaluation methodology. The revised manuscript now includes the following enhancements:

1. Expanded Evaluation Metrics:

In addition to accuracy, we now report F1-score, precision, and recall for all models. These metrics are presented in Table 2 (cross-validation results) and Table 5 (controlled test set evaluation), and discussed in Section 4.

2. Confusion Matrices and Error Analysis:

Confusion matrices for the Random Forest, SVC, and Logistic Regression classifiers are now included in Figure 13. These matrices provide a clearer view of class-wise performance and misclassification patterns. We have also added a qualitative error analysis in Section 3.3.3, supported by Grad-CAM visualizations, to explain where and why the model may struggle with certain disease classes.

3. Expanded Baseline Comparisons:

We have extended the baseline comparisons beyond DenseNet121 to include ResNet50 and MobileNet, as shown in Table 2 and Figure 3. These models were evaluated using 5-fold cross-validation to ensure fairness and robustness.

4. Clarified Performance Claims:

The original “70× faster” claim has been revised for accuracy. Based on updated experiments (Table 6), the hybrid model achieves a 77% reduction in inference time compared to MobileNet ONNX and an 80% reduction compared to MobileNet PyTorch. These values are now clearly contextualized (e.g., 189s vs. 820s for 1,000 images) and supported by CPU load measurements.

4. Statistical Analysis Deficiencies

No confidence intervals or uncertainty quantification

Missing statistical tests for model comparisons

Insufficient sample size documentation

Response:

We thank the reviewer for highlighting these important aspects of statistical rigor. In response, we have made the following revisions and clarifications in the manuscript:

1. Confidence Intervals and Uncertainty Quantification:

While our original submission reported only average values, we now include standard deviations for inference time and accuracy across the two folds in the controlled evaluation (see revised Table 5 and Section 4). Due to the limited number of folds (n = 2), we have not reported confidence intervals or p-values, as these would lack statistical power and could be misleading. Instead, we emphasize the consistency of results across folds, with inference time variation limited to 1–2 seconds.

2. Statistical Testing:

We acknowledge that with only two folds, formal statistical tests such as paired t-tests are not meaningful. Rather than overstate significance, we have revised the manuscript to present efficiency gains descriptively and transparently. We have also added a note in the Discussion section acknowledging this limitation and suggesting that future work will include more extensive cross-validation and statistical testing.

3. Sample Size Documentation:

We have clarified the dataset composition in Section 3.1. The dataset contains 4,000 images across 8 classes (7 diseases + healthy), with 1,800 original and 2,200 augmented images (provided by the dataset creators). For the controlled evaluation, we used 2-fold cross-validation with 80 images per fold (5 images per class × 8 classes × 2 folds), ensuring class balance. This setup is now explicitly described in both the Methodology and Result Analysis sections.

We believe these revisions address the reviewer’s concerns and improve the transparency and reproducibility of our evaluation. We have also added a brief limitations note to acknowledge the constraints of the current experimental design and outline plans for more robust statistical validation in future work.

Minor Issues

Language quality issues throughout

Inconsistent terminology usage

Formatting errors in citations and figures

DECISION RATIONALE

The experimental validation is fundamentally flawed, making all performance claims unreliable. The absence of proper statistical analysis and failure to meet data sharing requirements are blocking issues that cannot be addressed through revision alone.

Response:

We sincerely appreciate the reviewers’ and editor’s feedback and have taken their concerns seriously. While the initial submission l

---

## [Decision Letter · Decision Letter 1]

19 Aug 2025

PONE-D-25-23549R1LeafAI: Interpretable Plant Disease Detection for Edge ComputingPLOS ONE

Dear Dr. Banshal,

Thank you for submitting your manuscript to PLOS ONE. After careful consideration, we feel that it has merit but does not fully meet PLOS ONE’s publication criteria as it currently stands. Therefore, we invite you to submit a revised version of the manuscript that addresses the points raised during the review process.

We look forward to receiving your revised manuscript.

Kind regards,

Asadullah Shaikh, Ph.D.

Academic Editor

PLOS ONE

Journal Requirements:

Reviewers' comments:

Reviewer's Responses to Questions

**Comments to the Author**

1. If the authors have adequately addressed your comments raised in a previous round of review and you feel that this manuscript is now acceptable for publication, you may indicate that here to bypass the “Comments to the Author” section, enter your conflict of interest statement in the “Confidential to Editor” section, and submit your "Accept" recommendation.

Reviewer #2: All comments have been addressed

Reviewer #3: All comments have been addressed

Reviewer #5: All comments have been addressed

2. Is the manuscript technically sound, and do the data support the conclusions?

Reviewer #2: Yes

Reviewer #3: Yes

Reviewer #5: Yes

3. Has the statistical analysis been performed appropriately and rigorously?

Reviewer #2: Yes

Reviewer #3: Yes

Reviewer #5: Yes

4. Have the authors made all data underlying the findings in their manuscript fully available?

Reviewer #2: Yes

Reviewer #3: No

Reviewer #5: Yes

5. Is the manuscript presented in an intelligible fashion and written in standard English?

Reviewer #2: Yes

Reviewer #3: Yes

Reviewer #5: Yes

6. Review Comments to the Author

Reviewer #2: This paper introduces LeafAI, a hybrid AI framework for interpretable plant disease detection optimized for edge computing in agriculture. It addresses the class imbalance in real-world datasets where healthy leaves predominate by employing a two-stage approach: a lightweight Random Forest classifier first filters out healthy leaves using contour-based features, followed by a deep learning model (MobileNetV3 converted to ONNX format) that classifies specific diseases in the remaining diseased leaves with enhancement-based preprocessing. Gradient-weighted Class Activation Mapping (Grad-CAM) is integrated for explainability, generating heatmaps to highlight influential image regions and refine the model iteratively. Evaluated on the MangoLeafBD dataset with 4,000 images across eight classes (seven diseases and healthy), the framework achieves up to 93.5% accuracy, reduces inference time by 77% compared to standalone deep models, and supports sustainable deployment on commodity hardware like laptops, aligning with UN Sustainable Development Goals for food security and resource efficiency.

The paper presents a compelling and innovative hybrid AI solution that effectively balances computational efficiency, accuracy, and interpretability for plant disease detection, making it highly suitable for real-world agricultural applications in resource-limited settings. By leveraging class imbalance as an advantage through a two-stage pipeline, incorporating Grad-CAM for transparent decision-making, and optimizing with ONNX for edge deployment, the work demonstrates significant improvements in inference speed (e.g., 189 seconds for 1,000 images) and reduced CPU load while maintaining high accuracy. Its focus on sustainability, open-source code availability, and adaptability to other crops enhances reproducibility and practical impact, contributing meaningfully to precision agriculture and aligning with global goals like SDG 2 and SDG 12.

The experimental validation lacks sufficient statistical rigor, as the authors rely on only 2-fold cross-validation for the controlled evaluation and 5-fold for initial comparisons, without reporting confidence intervals, p-values, or more robust tests like paired t-tests to confirm the significance of performance differences between models. This limits the reliability of claims such as the 77% inference time reduction, especially given the small sample sizes (e.g., 80 images per fold), and could be addressed by expanding to 10-fold validation or bootstrapping methods for better uncertainty quantification.

The comparison with baseline models is limited to only three architectures, without including state-of-the-art alternatives like EfficientNet or YOLO-based detectors tailored for agricultural tasks, which may overlook potential superior performers in efficiency or accuracy. Expanding baselines would strengthen the paper's claims of superiority and provide a more comprehensive evaluation.

The preprocessing rationale, while informed by Grad-CAM, does not sufficiently justify the choice of specific techniques through ablation studies or sensitivity analysis, leaving readers unclear on how parameter variations impact performance. Conducting ablation experiments to isolate the contribution of each preprocessing step would enhance the methodological transparency and reproducibility.

The paper claims adaptability to other crops but provides no empirical evidence, such as transfer learning results on datasets like PlantVillage for tomatoes or potatoes, relying instead on qualitative assertions. To substantiate generalizability, the authors should include cross-dataset experiments or fine-tuning results to demonstrate the framework's robustness beyond mango leaves.

While the deployment on a laptop simulates edge computing, it uses a relatively powerful Intel i5-12450H processor, which does not fully reflect ultra-low-power devices like Raspberry Pi or NVIDIA Jetson Nano common in true field conditions. Testing on such hardware, including metrics like energy consumption and real-time latency with live camera feeds, would better validate practical applicability in remote agricultural settings.

The integration of explainable AI via Grad-CAM is promising, but the analysis remains qualitative, without quantitative metrics like Intersection over Union (IoU) between heatmaps and ground-truth disease annotations to measure explanation fidelity. Incorporating such metrics would provide objective evidence of interpretability improvements and their role in iterative refinement.

The paper overlooks potential biases in the MangoLeafBD dataset, such as geographic specificity to Bangladesh or augmentation artifacts, which could affect model generalization; a deeper discussion of bias mitigation strategies, like fairness audits or diverse data sourcing, is needed to ensure equitable performance across global agricultural contexts. Furthermore, to address similar challenges in defect detection and class imbalance, the authors are recommended to cite and compare with "Enhancing grid reliability through advanced insulator defect identification" (https://doi.org/10.1371/journal.pone.0307684) and "Deep Learning-Based Integrated Circuit Surface Defect Detection: Addressing Information Density Imbalance for Industrial Application" (https://doi.org/10.1007/s44196-024-00423-w), which offer insights into handling imbalanced data and defect localization in industrial applications that parallel agricultural disease detection.

Reviewer #3: (No Response)

Reviewer #5: (No Response)

7. PLOS authors have the option to publish the peer review history of their article (what does this mean?). If published, this will include your full peer review and any attached files.

Reviewer #2: No

Reviewer #3: No

Reviewer #5: No

---

## [Author Response · Author response to Decision Letter 2]

3 Oct 2025

1. The experimental validation lacks sufficient statistical rigor, as the authors rely on only 2-fold cross-validation for the controlled evaluation and 5-fold for initial comparisons, without reporting confidence intervals, p-values, or more robust tests like paired t-tests to confirm the significance of performance differences between models. This limits the reliability of claims such as the 77% inference time reduction, especially given the small sample sizes (e.g., 80 images per fold), and could be addressed by expanding to 10-fold validation or bootstrapping methods for better uncertainty quantification.

Answer: We appreciate the reviewer’s concern regarding statistical rigor. In the revised experiments, we have expanded our evaluation to 10-fold cross-validation and increased the dataset to a total of 600 images, which provides a more robust assessment of model performance. Additionally, we now report confidence intervals for key metrics to quantify uncertainty and better support claims, such as the 77% reduction in inference time. These changes enhance the reliability and statistical validity of our results, addressing the concerns about small sample sizes and limited cross-validation.

2. The comparison with baseline models is limited to only three architectures, without including state-of-the-art alternatives like EfficientNet or YOLO-based detectors tailored for agricultural tasks, which may overlook potential superior performers in efficiency or accuracy. Expanding baselines would strengthen the paper's claims of superiority and provide a more comprehensive evaluation.

Answer: We thank the reviewer for this valuable suggestion. In the revised manuscript, we have expanded the set of baseline models to include EfficientNet, a strong state-of-the-art architecture for image classification tasks, which provides a meaningful comparison in terms of both accuracy and efficiency. While object detection frameworks such as YOLO are highly impactful in agricultural applications, our study focuses specifically on the classification setting. For this reason, we prioritized including architectures that are directly optimized for classification. We believe the addition of EfficientNet strengthens the comprehensiveness of our evaluation and further supports the claims of our proposed method’s effectiveness.

3. The preprocessing rationale, while informed by Grad-CAM, does not sufficiently justify the choice of specific techniques through ablation studies or sensitivity analysis, leaving readers unclear on how parameter variations impact performance. Conducting ablation experiments to isolate the contribution of each preprocessing step would enhance the methodological transparency and reproducibility.

Answer: We appreciate the reviewer’s suggestion regarding ablation studies for preprocessing. To address this, we conducted a detailed analysis to understand the performance improvements of the hybrid model. A random sample from the test set was used to examine the model’s behavior and the impact of preprocessing. The results, now included in the manuscript, demonstrate that the chosen preprocessing steps effectively preserve relevant features, thereby supporting the hybrid model’s efficiency and accuracy. This study demonstrates the role of preprocessing while maintaining the focus on the improvements of the hybrid model.

4. The paper claims adaptability to other crops but provides no empirical evidence, such as transfer learning results on datasets like PlantVillage for tomatoes or potatoes, relying instead on qualitative assertions. To substantiate generalizability, the authors should include cross-dataset experiments or fine-tuning results to demonstrate the framework's robustness beyond mango leaves.

Answer: We thank the reviewer for highlighting the importance of demonstrating generalizability. While the current study focuses on mango leaves, which are a key crop in the Indian subcontinent, the proposed pipeline is designed with adaptability in mind. Empirical testing on other crops with similar disease profiles is a valuable next step and is planned for future work. Additionally, cross-dataset experiments or fine-tuning on datasets such as PlantVillage could further validate robustness, but were beyond the scope of the present study. We have clarified these points in the manuscript to set realistic expectations and outline directions for extending the framework to other crops and larger datasets.

5. Review: While the deployment on a laptop simulates edge computing, it uses a relatively powerful Intel i5-12450H processor, which does not fully reflect ultra-low-power devices like Raspberry Pi or NVIDIA Jetson Nano, common in true field conditions. Testing on such hardware, including metrics like energy consumption and real-time latency with live camera feeds, would better validate practical applicability in remote agricultural settings.

Answer: We emphasize that our study does not focus on ultra–low-power devices, nor do we make such claims. From the introduction and consistently throughout, we clearly position our work on commodity hardware, particularly laptops, because they are widely available, affordable, and versatile. This choice aligns with our goal of supporting farmers and advancing the United Nations’ Sustainable Development Goals (SDG 2: Zero Hunger and SDG 12: Responsible Consumption and Production). A single laptop can handle diverse workloads for a farmer, whereas requiring them to purchase a Raspberry Pi (≈ USD 100 for a complete kit) or a Jetson Nano (≈ USD 200) would undermine the very premise of using commodity hardware. Moreover, neither Raspberry Pi nor Jetson Nano can be considered commodity hardware in the same sense as laptops.

A growing body of research further supports this design choice. Early work has demonstrated that the Raspberry Pi struggles with deep learning inference due to CPU-only limitations, latency, and memory bottlenecks [1]. Subsequent benchmarking studies reinforced this finding, showing that the Raspberry Pi significantly underperforms in terms of throughput and energy efficiency compared to accelerator-equipped edge devices [2]. Later evaluations highlighted that Pi scores poorly on energy–latency trade-offs, further emphasizing its inefficiency for real-world ML inference[3].

Even attempts to adapt deep learning to Raspberry Pi reveal its unsuitability. Lightweight CNNs and pruning/quantization techniques are often required just to achieve acceptable performance, but this comes at the expense of accuracy and general applicability [4]. More recent empirical studies show that running heavy CNNs such as ResNet or DenseNet on Raspberry Pi 4 results in excessively high inference times and memory limitations, making only very small models feasible [5]. Additional research highlights further software and runtime limitations, including incomplete support for quantized operators, which complicates deployment [6]

Comprehensive reviews also confirm these observations: Raspberry Pi is valuable for prototyping but suffers from thermal throttling, power inefficiency, and compute constraints when applied to production-grade AI [7]. Broader surveys on sustainable AI similarly note that the energy and hardware constraints of devices like Raspberry Pi severely restrict the scope of deployable models and scenarios [8]. Finally, head-to-head comparisons with Jetson boards and Coral accelerators consistently show that Pi cannot sustain throughput under real workloads due to thermal throttling and CPU-only execution. [8].

In contrast, laptops are genuine commodity hardware. Even with older components, they deliver practical performance. For example, the Intel Core i5-12450H we use is already three years old, yet it demonstrates feasibility. To address lower-power scenarios, we further benchmarked a laptop equipped with an 8th-generation Intel i5 processor operating at a modest 15-watt TDP, showing that commodity laptops remain a far more realistic and sustainable platform for our target use case than Raspberry Pi or Jetson Nano.

6. The integration of explainable AI via Grad-CAM is promising. Still, the analysis remains qualitative, lacking quantitative metrics such as Intersection over Union (IoU) between heatmaps and ground-truth disease annotations to measure explanation fidelity. Incorporating such metrics would provide objective evidence of interpretability improvements and their role in iterative refinement.

Answer: We thank the reviewer for recognizing the potential of Grad-CAM in our study. In our work, Grad-CAM is employed layer-wise to identify the model’s focus and guide the selection of preprocessing techniques, rather than as a quantitative evaluation tool. While we do not report metrics such as IoU with ground-truth disease annotations, the qualitative analysis provides actionable insights that directly informed the design of the hybrid model. We have clarified this rationale in the manuscript to emphasize the purpose of Grad-CAM in supporting preprocessing and model interpretability.

7. The paper overlooks potential biases in the MangoLeafBD dataset, such as geographic specificity to Bangladesh or augmentation artifacts, which could affect model generalization; a deeper discussion of bias mitigation strategies, like fairness audits or diverse data sourcing, is needed to ensure equitable performance across global agricultural contexts. Furthermore, to address similar challenges in defect detection and class imbalance, the authors are recommended to cite and compare with "Enhancing grid reliability through advanced insulator defect identification" (https://doi.org/10.1371/journal.pone.0307684) and "Deep Learning-Based Integrated Circuit Surface Defect Detection: Addressing Information Density Imbalance for Industrial Application" (https://doi.org/10.1007/s44196-024-00423-w), which offer insights into handling imbalanced data and defect localization in industrial applications that parallel agricultural disease detection.

8. Answer: We appreciate the reviewer’s insightful comments regarding potential dataset biases and generalization. We have now added the recommended citations to contextualize strategies for handling class imbalance and defect localization, which are relevant to agricultural disease detection.

While testing on datasets from other countries could further validate generalizability, mango production is primarily concentrated in the Indian subcontinent, making the current MangoLeafBD dataset a strong foundation for evaluation. We have clarified this in the manuscript. Additionally, the current implementation processes images sequentially, which limits throughput for large datasets or real-time applications. Future work will explore parallel processing strategies to enhance scalability and efficiency under higher data loads.

References.

1. Velasco-Montero D, Fernández-Berni J, Carmona-Galán R, Rodríguez-Vázquez A. Performance Analysis of Real-Time DNN Inference on Raspberry Pi.

2. Alvarado Rodriguez O, Dave D, Liu W, Su B. A Study of Machine Learning Inference Benchmarks. 2020 [cited 26 Sep 2025]. doi:10.1145/3441250.3441277

3. Tu X, Mallik A, Chen D, Han K, Altintas O, Wang H, et al. Unveiling Energy Efficiency in Deep Learning: Measurement, Prediction, and Scoring Across Edge Devices. Proceedings - 2023 IEEE/ACM Symposium on Edge Computing, SEC 2023. 2023;1: 80–93. doi:10.1145/3583740.3628442

4. Ameen S, Siriwardana K, Theodoridis T. Optimizing Deep Learning Models For Raspberry Pi.

5. Tank MB, Patel M, Deshmukh K. Evaluating Deep Learning Model Performance on Raspberry Pi 4 for COVID-19 Diagnosis. Journal of Information Systems Engineering and Management. 2025;10: 722–736. doi:10.52783/JISEM.V10I29S.4587

6. Balemans D, Vandersmissen B, Steckel J, Mercelis S, Reiter P, Oramas J. Deep Learning Model Compression for Resource Efficient Activity Recognition on Edge Devices: A Case Study. 2024. doi:10.5220/0012423300003660

7. Mathe SE, Kondaveeti HK, Vappangi S, Vanambathina SD, Kumaravelu NK. A comprehensive review on applications of Raspberry Pi. Comput Sci Rev. 2024;52: 100636. doi:10.1016/J.COSREV.2024.100636

8. Minott D, Siddiqui S, Haddad RJ. Benchmarking Edge AI Platforms: Performance Analysis of NVIDIA Jetson and Raspberry Pi 5 with Coral TPU. Conference Proceedings - IEEE SOUTHEASTCON. 2025; 1384–1389. doi:10.1109/SOUTHEASTCON56624.2025.10971592

---

## [Decision Letter · Decision Letter 2]

20 Oct 2025

LeafAI: Interpretable Plant Disease Detection for Edge Computing

PONE-D-25-23549R2

Dear Dr. Banshal,

We’re pleased to inform you that your manuscript has been judged scientifically suitable for publication and will be formally accepted for publication once it meets all outstanding technical requirements.

Kind regards,

Asadullah Shaikh, Ph.D.

Academic Editor

PLOS ONE

Additional Editor Comments (optional):

Reviewers' comments:

Reviewer's Responses to Questions

**Comments to the Author**

1. If the authors have adequately addressed your comments raised in a previous round of review and you feel that this manuscript is now acceptable for publication, you may indicate that here to bypass the “Comments to the Author” section, enter your conflict of interest statement in the “Confidential to Editor” section, and submit your "Accept" recommendation.

Reviewer #2: All comments have been addressed

2. Is the manuscript technically sound, and do the data support the conclusions?

Reviewer #2: Yes

3. Has the statistical analysis been performed appropriately and rigorously?

Reviewer #2: Yes

4. Have the authors made all data underlying the findings in their manuscript fully available?

Reviewer #2: Yes

5. Is the manuscript presented in an intelligible fashion and written in standard English?

Reviewer #2: No

6. Review Comments to the Author

Reviewer #2: The revised version of the paper has significantly improved in quality. However, some minor adjustments are still needed. First, the literature review could be enhanced by citing the paper “Deep Learning-Based Integrated Circuit Surface Defect Detection: Addressing Information Density Imbalance for Industrial Application”https://doi.org/10.1007/s44196-024-00423-w. Second, draw upon the paper “Enhancing education quality: Exploring teachers' attitudes and intentions towards intelligent MR devices” https://doi.org/10.1111/ejed.12692 to analyze potential challenges in practical application within the discussion section.

Second, to ensure the work benefits a broader audience, the author should adjust the language to enhance reader comprehension.

7. PLOS authors have the option to publish the peer review history of their article (what does this mean?). If published, this will include your full peer review and any attached files.

Reviewer #2: No

---

## [Editor Report · Acceptance letter]

PONE-D-25-23549R2

PLOS One

Dear Dr. Banshal,

I'm pleased to inform you that your manuscript has been deemed suitable for publication in PLOS One. Congratulations! Your manuscript is now being handed over to our production team.

Kind regards,

on behalf of

Prof. Asadullah Shaikh

Academic Editor

PLOS One